# Zika virus targets human trophoblast stem cells and prevents syncytialization in placental trophoblast organoids

Hao Wu[1,2,3,4,9], Xing-Yao Huang[5,9], Meng-Xu Sun[5,9], Yue Wang[1,2,3,4,9], Hang-Yu Zhou[6,9], Ying Tian[5,7,9], Beijia He[1,2,3,4], Kai Li[5], De-Yu Li[5], Ai-Ping Wu ®[6], Hongmei Wang ®[1,2,3,4] ✉ & Cheng-Feng Qin ®[5,8] ✉

Zika virus (ZIKV) infection during pregnancy threatens pregnancy and fetal health. However, the infectivity and pathological effects of ZIKV on placental trophoblast progenitor cells in early human embryos remain largely unknown. Here, using human trophoblast stem cells (hTSCs), we demonstrated that hTSCs were permissive to ZIKV infection, and resistance to ZIKV increased with hTSC differentiation. Combining gene knockout and transcriptome analysis, we demonstrated that the intrinsic expression of *AXL* and *TIM-1*, and the absence of potent interferon (IFN)-stimulated genes (ISGs) and IFNs contributed to the high sensitivity of hTSCs to ZIKV. Furthermore, using our newly developed hTSC-derived trophoblast organoid (hTSC-organoid), we demonstrated that ZIKV infection disrupted the structure of mature hTSC-organoids and inhibited syncytialization. Single-cell RNA sequencing (scRNA-seq) further demonstrated that ZIKV infection of hTSC-organoids disrupted the stemness of hTSCs and the proliferation of cytotrophoblast cells (CTBs) and probably led to a preeclampsia (PE) phenotype. Overall, our results clearly demonstrate that hTSCs represent the major target cells of ZIKV, and a reduced syncytialization may result from ZIKV infection of early developing placenta. These findings deepen our understanding of the characteristics and consequences of ZIKV infection of hTSCs in early human embryos.

Zika virus (ZIKV), a positive-strand RNA arbovirus of the *Flaviviridae* family, has been reported to be autochthonously transmitted in more than 80 countries worldwide[1]. For adults, most infected individuals are asymptomatic or exhibit mild flu-like symptoms[2]. However, ZIKV infection in pregnant women frequently cause severe congenital anomalies, including neurologic and ocular abnormalities, fetal growth restriction, stillbirth, and perinatal death. In particular, a dramatic increase in neonatal microcephaly cases in Brazil led to the World Health Organization (WHO) issuing a public health emergency of international concern[1]. Moreover, viral RNA and protein have been

[1]State Key Laboratory of Stem Cell and Reproductive Biology, Institute of Zoology, Chinese Academy of Sciences, Beijing 100101, China. [2]Institute for Stem Cell and Regeneration, Chinese Academy of Sciences, Beijing 100101, China. [3]Beijing Institute for Stem Cell and Regenerative Medicine, Beijing 100101, China. [4]University of Chinese Academy of Sciences, Beijing 100049, China. [5]State Key Laboratory of Pathogen and Biosecurity, Beijing Institute of Microbiology and Epidemiology, Academy of Military Medical Sciences, Beijing 100071, China. [6]State Key Laboratory of Common Mechanism Research for Major Diseases, Suzhou Institute of Systems Medicine, Chinese Academy of Medical Sciences & Peking Union Medical College, Suzhou 215123, China. [7]School of Basic Medical Sciences, Anhui Medical University, Hefei 230032, China. [8]Research Unit of Discovery and Tracing of Natural Focus Diseases, Chinese Academy of Medical Sciences, Beijing 100071, China. [9]These authors contributed equally: Hao Wu, Xing-Yao Huang, Meng-Xu Sun, Yue Wang, Hang-Yu Zhou, Ying Tian. ✉e-mail: wanghm@ioz.ac.cn; qincf@bmi.ac.cn

detected in placental tissues, amniotic fluid, and fetal brain, suggesting maternal-to-fetal transmission of ZIKV via the placenta[3–7].

The placenta develops from the trophectoderm (TE), which forms in preimplantation human embryos at ~5 days post-fertilization (dpf)[8]. Following implantation, the TE fuses and invades into the uterine surface epithelium. Throughout pregnancy, the placenta functions as a physical and immunological barrier against the hematogenous transmission of virus from mother to fetus, and serves as a vital functional organ to support fetal development[9]. Placental villi are the functional units of the placenta, and are composed of floating and anchoring villi. Specialized fetal-derived villous trophoblast cells in chorionic villi mediate the interaction between the mother and fetus. Trophoblast cells include three major cell subpopulations cytotrophoblast cell (CTB), syncytiotrophoblast (STB), and extravillous trophoblast cell (EVT). CTB is a proliferative population and can give rise to multinucleated STB and invasive EVT. The STB overlies the mononucleated CTB in the villi, and is in direct contact with maternal blood. The STB layer is considered to be highly resistant to numerous pathogenic infections[10]. The EVT at the tip of the anchoring villi invades the maternal decidua, which establishes physical connections between the mother and fetus.

Multiple laboratory cellular models of ZIKV infection have been developed to study its trans-placenta transmission, mainly including primary human trophoblast cells, placental chorionic villi explants, human embryonic stem cell (hESC)-derived trophoblast-like cells, and choriocarcinoma cell lines, such as BeWo, JEG-3, and JAR[11–16]. An immunocompromised mouse-based ZIKV infection model also provides a powerful tool to study the in vivo pathogenesis of ZIKV infection[17–19]. The lack of immune responses in choriocarcinoma cells limits their use in studying the mechanism of antiviral infection in human trophoblast cells. The hESC-derived trophoblast-like cell-based ZIKV infection model has provided advanced insights into ZIKV infection of the human placenta. Sheriden et al. indicated the vulnerability of hESC-derived trophoblast-like cells to ZIKV infection[16], while some studies pointed a possible difference between hESC-derived trophoblast-like cells and primary trophoblast cells[20]. Tan et al. showed that the TE cells of preimplantation human embryos can be infected by ZIKV[21]. Using an in vitro cultured primary cell-based ZIKV infection model, the CTB and invasive CTB in cultured chorionic villous explants from first-trimester placentas and the CTB from second- and third-trimester placentas were shown to be infected by ZIKV[22]. STB in villous explants from the first-trimester placenta was resistant to ZIKV[11]. Distinct susceptibility to ZIKV of different types of trophoblast cells was shown in primary cells. However, what leads to different susceptibilities to ZIKV between CTB, STB and EVT has not been studied. Thus, more stable, flexible and physiological cell models are needed to explore the characteristics and mechanisms of ZIKV infection of different types of human trophoblast cells.

Recently, human trophoblast stem cells (hTSCs) have been derived from the TE of blastocysts, as well as from the first-trimester placentas, and are considered to be the cells that best meet the criteria for human trophoblast cells[23–25]. The hTSCs can be induced to differentiate into STB and EVT in vitro. Furthermore, the construction of primary cell- or stem cell-based three-dimensional (3D) placental trophoblast organoids provides transformative models for investigating the interactions between trophoblast cells and maternal environment[26–28]. Trophoblast organoids also serve as a valuable model for exploring the placental accessibility to pathogens[28].

Intrinsic expression of certain host molecules and interferon (IFN)-stimulated genes (ISGs) in target cells determines the susceptibility to virus infection. The expression of ISGs confers cellular resistance to viruses. IFN constitutes a defense against viruses by affecting the production of ISGs, which act differently in different stages of the viral life cycle, from entry, replication and assembly to release. Several intrinsically expressed ISGs have been reported to directly inhibit virus

infection, such as *IRFs*, *IFITs*, *IFITMs*, *ISG15*, *OASL*, *RIG-1* and *IRFs*[29]. Bayer et al. demonstrated that the type III IFN produced by placental trophoblast cells conferred resistance of JEG-3 to ZIKV infection[12]. However, the intrinsic ISG profiles in different types of trophoblast cells remain unclear. The entry of flaviviruses into host cells mainly depends on endocytosis, which initiates from the interaction between flaviviruses and cellular host factors. *TAM* and *TIM* family genes were identified as potential host factors mediating flavivirus infection of human cells by cDNA screening[30]. However, *TAM* and *TIM* family genes showed different functions in flavivirus infection of cells from different tissues.

In this work, using hTSCs, we establish ZIKV infection in hTSCs and hTSC-derived STB[TS] and EVT[TS]. We demonstrate that hTSCs are permissive to ZIKV infection and that the susceptibility to ZIKV decreased with hTSC differentiation. The expression levels of *AXL* and *TIM-1* contribute to the susceptibility of hTSC-derived trophoblast cells to ZIKV infection. The hTSCs lack intrinsic expression of representative antiviral ISGs and IFNs. Furthermore, we construct a 3D human placental trophoblast organoid using hTSCs (hTSC-organoid) to model the physiological development of human placental trophoblast, and establish ZIKV infection. Through this model, we demonstrate that ZIKV infection disrupts the structure of mature hTSC-organoids and inhibits syncytialization. Single-cell RNA sequencing (scRNA-seq) further demonstrates that ZIKV infection of hTSC-organoids may disrupt the stemness of hTSCs and the proliferation of CTBs and may lead to a preeclampsia (PE) phenotype.

## Results

### hTSCs, STB[TS] and EVT[TS] have distinct vulnerabilities to ZIKV infection

The hTSCs possess the characteristics of mononucleated CTB, which can give rise to STB[TS] expressing CGB and EVT[TS] expressing HLA-G[24,25]. In this study, using hTSCs isolated from the human blastocyst, we investigated the infectivity and pathogenicity of ZIKV to hTSCs and hTSC-differentiated STB[TS] and EVT[TS].

First, to characterize the susceptibility of hTSCs, STB[TS] and EVT[TS] to ZIKV infection and replication, hTSCs, STB[TS] and EVT[TS] were plated at the same confluence in 24-well plates and inoculated with ZIKV (GZ01, isolated from a patient returned from Venezuela in 2016) at a multiplicity of infection (MOI) of 0.1. The supernatants of hTSCs, STB[TS], and EVT[TS] were collected at 12, 24 and 48 hours post infection (hpi), and viral RNA was detected by qPCR. As shown in Fig. 1a, the growth of viral RNA in the supernatants of all three types of trophoblast cells was observed. In hTSCs, the viral RNA in the supernatants exhibited an approximate 13.5-fold increase from 12 to 48 hpi, suggesting that hTSCs supported ZIKV infection and replication. In STB[TS] and EVT[TS], the growth rate of ZIKV RNA was significantly lower than that in hTSCs (Fig. 1a).

Next, to demonstrate the viral entry into cells, we performed immunofluorescence (IF) staining for ZIKV envelope (E) protein, which is responsible for virus entry, in the hTSCs, STB[TS] and EVT[TS] infected with ZIKV at an MOI of 0.1. At 24 hpi, clear intracellular ZIKV E signals were observed in hTSCs (Fig. 1b). In contrast, ZIKV E signals were observed only in a few views in STB[TS] and no ZIKV E signal was seen in EVT[TS] at all (Fig. 1c and d). At 48 hpi, the proportion of intracellular ZIKV E signals in hTSCs, STB[TS] and EVT[TS] increased, which was consistent with the trend of viral RNA in the supernatants (Fig. 1b–d). We counted the proportion of ZIKV E-positive cells in three types of trophoblast cells in ten randomly selected fields, which also showed a decrease in the proportion of infected cells with hTSC differentiation and an increase in infected cells over time (Fig. 1e, Table S1 and S2). Of note, in EVT[TS], only very slight viral signals could be found in few views at 48 hpi, with an average of 1.57% infected cells (Fig. 1e, Table S2).

To further evaluate the ability of trophoblast cells to produce infectious viral particles, we detected the viral titers in the

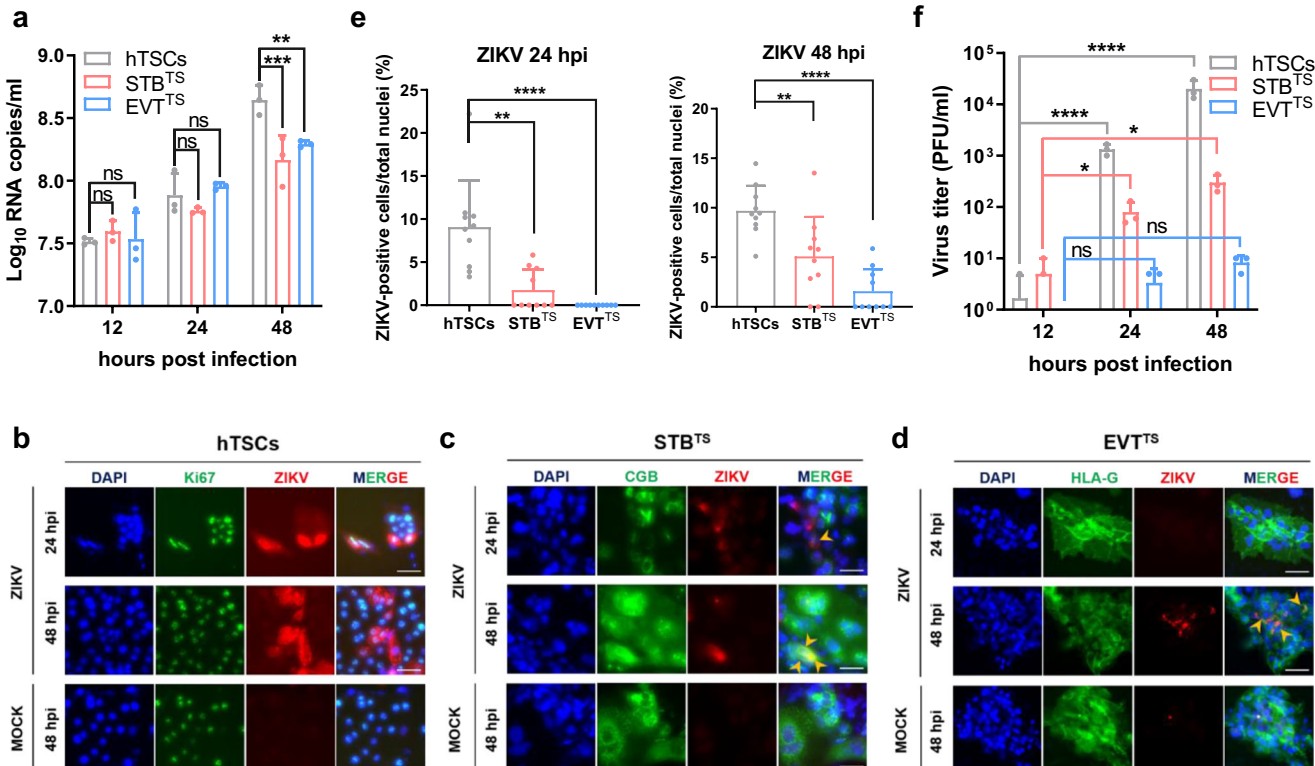

**Fig. 1 | hTSCs, STB^TS and EVT^TS have distinct vulnerabilities to ZIKV infection.**
**a** Quantification of ZIKV RNA in the supernatants of hTSCs, STB^TS and EVT^TS at 12, 24 and 48 hours post infection. The hTSCs, STB^TS and EVT^TS were infected with ZIKV at an MOI of 0.1. Two-way ANOVA analysis was used for statistical analysis of significance. $n = 3$ independent experiments. **, $p = 0.0069$. ***, $p = 0.0004$. ns, no significance. **b**–**d** Immunofluorescence staining for Ki67 (a marker of proliferative hTSCs), CGB (a marker of STB^TS), HLA-G (a marker of EVT^TS) and ZIKV E protein in hTSCs (panel **b**), STB^TS (panel **c**) and EVT^TS (panel **d**). Nuclei were stained with DAPI. The hTSCs, STB^TS and EVT^TS were infected with ZIKV at an MOI of 0.1 and were analyzed at 24 and 48 hours post infection. The yellow arrow heads indicated the positive intracellular ZIKV E signals in STB^TS and EVT^TS. Scale bars: 50 μm.

**e** Statistical analysis of the proportion of ZIKV E-positive cells in the hTSCs, STB^TS and EVT^TS infected with ZIKV at 24 (left panel) and 48 (right panel) hours post infection. Two-tailed unpaired t test was used for statistical analysis of significance. $n = 10$ random views. Left panel, **, $p = 0.001$. ****, $p < 0.0001$. Right panel, **, $p = 0.0066$. ****, $p < 0.0001$. **f** Analysis of viral titer in the supernatants of hTSCs, STB^TS and EVT^TS at 12, 24 and 48 hours post infection by plaque-forming assay. The hTSCs, STB^TS and EVT^TS were infected with ZIKV at an MOI of 0.1. Two-way ANOVA was used for statistical analysis of significance. $n = 3$ independent experiments. hTSCs, ****, $p < 0.0001$. STB^TS, 12 hpi vs 24 hpi, *, $p = 0.0399$. 12 hpi vs 48 hpi, *, $p = 0.0261$. ns, no significance. Data in this figure are shown as the mean ± s.d.

supernatants of ZIKV-infected hTSCs, STB^TS and EVT^TS by plaque-forming assay, which showed increasing viral particles over time, indicating an active viral replication in hTSCs, STB^TS and EVT^TS (Fig. 1f). To further verify the different sensitivities of different trophoblast cell types to ZIKV, we detected the growth of viral RNA in another blastocyst-derived hTSC strain (BT1, gifted by Arima lab), and STB^BT1 and EVT^BT1 exhibited less susceptibility to ZIKV than hTSCs (BT1) (Fig. S1A). To explore whether hTSCs have broad susceptibility to ZIKV, we infected hTSCs, STB^TS and EVT^TS with another ZIKV strain (FSS 13025) at an MOI of 0.1 and detected the viral RNA in the supernatants at 12, 24 and 48 hpi. FSS 13025 showed similar infection characteristics to GZ01 in hTSC-derived trophoblast cells (Fig. S1B). Taken together, these results demonstrated that the hTSCs isolated from human blastocyst were permissive to ZIKV infection and proliferation, and the susceptibility to ZIKV decreased after differentiation.

### AXL and TIM-1 facilitate ZIKV infection in hTSCs

*TAM* family and *TIM* family genes, mainly including *AXL*, *TYRO3*, *MERTK* and *TIM-1*, were suggested as putative host factors in flavivirus infection of human host cells[30–33]. To explore the role of these potential host factors in ZIKV infection of hTSCs, STB^TS and EVT^TS, we compared the expression of *AXL*, *TYRO3*, *MERTK* and *TIM-1* in hTSCs, STB^TS and EVT^TS by qRT-PCR. *AXL*, *MERTK* and *TIM-1* showed much higher expression in hTSCs than in STB^TS and EVT^TS (Fig. 2a). *TYRO3* was highly expressed in hTSCs and was further up-regulated after differentiation (Fig. 2a).

Then, to explore whether the differential expression of the potential host factors determined the differential susceptibility to ZIKV infection, we performed a double allele mutation of the coding sequences of *AXL*, *TYRO3*, *MERTK* and *TIM-1* in hTSCs using CRISPR/Cas9, and successful code-shifting mutations of these genes in hTSCs were demonstrated by Sanger sequencing and qRT-PCR (Figure S2). We inoculated wildtype (WT) and host factor knockout hTSCs with ZIKV at an MOI of 0.1, and performed IF staining for ZIKV E protein at 48 hpi. As shown in Fig. 2b, significantly attenuated ZIKV infection was found in *AXL*⁻/⁻ and *TIM-1*⁻/⁻ hTSCs compared to WT hTSCs. Moreover, the replication of ZIKV RNA was also reduced in *AXL*⁻/⁻ and *TIM-1*⁻/⁻ hTSCs, indicating the important roles of *AXL* and *TIM-1* in ZIKV infection of hTSCs (Fig. 2c). In contrast, knockout of *TYRO3* or *MERTK* showed no significant effect on ZIKV infection of hTSCs (Figure S3). These results suggested that the knockout of *AXL* and *TIM-1* in *TAM* and *TIM* family genes could inhibit ZIKV infection of hTSCs.

To examine whether *AXL* or *TIM-1* could endow EVT permissiveness to ZIKV infection, we constructed doxycycline (DOX)-inducible hTSCs overexpressing *AXL* (TRE_AXL) or *TIM-1* (TRE_TIM-1) (Figure S4A and S4B). We differentiated the indicated hTSCs into EVT^TS (TRE_AXL-EVT^TS and TRE_TIM-1-EVT^TS) and then treated them with DOX for 48 hours before performing ZIKV infection. *AXL* and *TIM-1* were successfully overexpressed in hTSCs and EVT^TS after treatment with 5 μM DOX (Figure S4C and S4D). We inoculated TRE_AXL-EVT^TS and TRE_TIM-1-EVT^TS with ZIKV at an MOI of 0.1. As

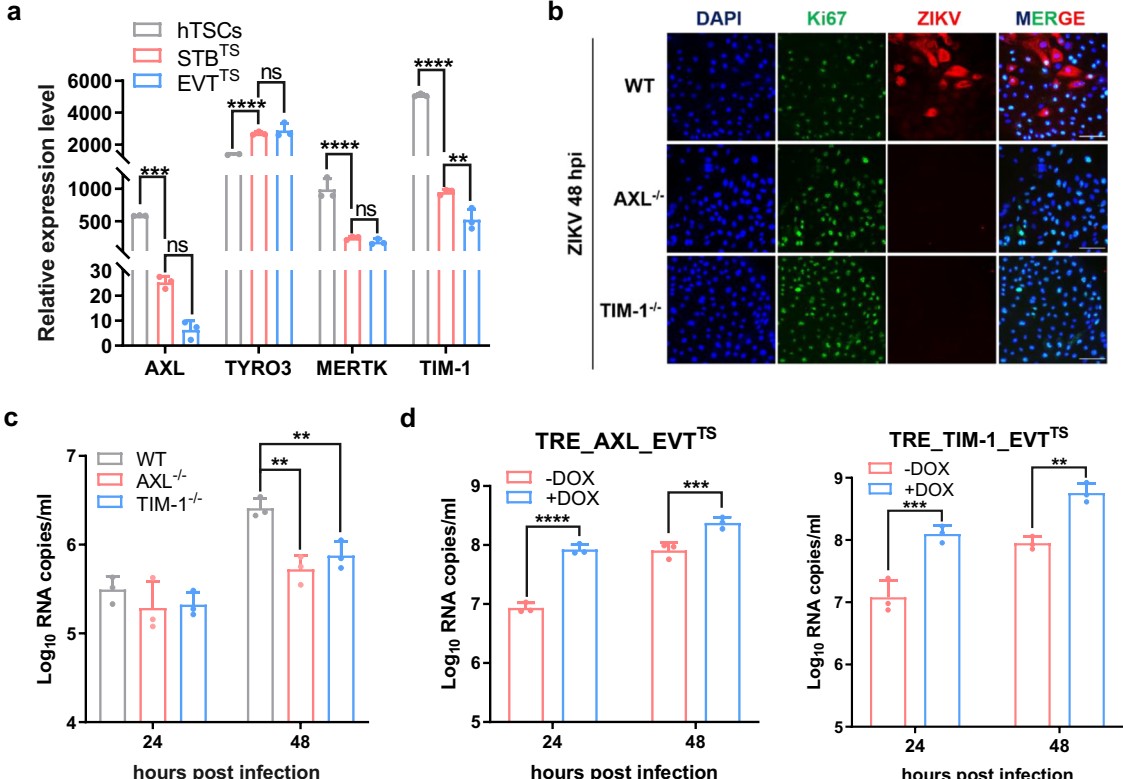

**Fig. 2 | Expression of AXL and TIM-1 facilitates ZIKV infection in hTSCs.**
**a** Relative expression levels of the putative host factors AXL, TYRO3, MERTK and TIM-1 in hTSCs, STB[TS] and EVT[TS]. Two-way ANOVA analysis was used for statistical analysis of significance. $n = 3$ independent experiments. AXL, ***, $p = 0.0004$. TYRO3, ****, $p < 0.0001$. MERTK, ****, $p < 0.0001$. TIM-1, **, $p < 0.0064$. ****, $p < 0.0001$. ns, no significance. **b** Representative immunofluorescence images for Ki67 and ZIKV E protein in ZIKV-infected WT, AXL[-/-] and TIM-1[-/-] hTSCs. Nuclei were stained with DAPI. WT, AXL[-/-] and TIM-1[-/-] hTSCs were exposed to ZIKV at an MOI of 0.1, and analyzed at 48 hours post infection. Scale bars: 100 μm. **c** Quantification of viral RNA in the supernatants of ZIKV-infected WT, AXL[-/-] and TIM-1[-/-]hTSCs at 24

and 48 hours post infection. Two-way ANOVA analysis was used for statistical analysis of significance. $n = 3$ independent experiments. 48 hpi, WT vs AXL[-/-], **, $p = 0.0014$. WT vs TIM-1[-/-], **, $p = 0.0089$. **d** Quantification of viral RNA in the supernatants of ZIKV-infected EVT[TS] with DOX-inducible ectopic expression of AXL (TRE_AXL_EVT[TS], left panel) and TIM-1 (TRE_TIM-1_EVT[TS], right panel) at 24 and 48 hours post infection. EVT[TS] was pretreated with 5 μM DOX for 48 hours and infected with ZIKV at an MOI of 0.1. Two-way ANOVA analysis was used for statistical analysis of significance. $n = 3$ independent experiments. Left panel, ***, $p = 0.0009$. ****, $p < 0.0001$. Right panel, **, $p = 0.0011$. ***, $p = 0.0002$. Data in this figure are shown as the mean ± s.d.

expected, overexpression of either *AXL* or *TIM-1* significantly enhanced ZIKV infection in EVT[TS] (Fig. 2d). In contrast, DOX treatment in WT-EVT[TS] had no effect on ZIKV infection (Figure S4E). Taken together, these results demonstrated the essential roles of *AXL* and *TIM-1* in the susceptibility of hTSC-derived trophoblast cells to ZIKV infection.

## Lack of intrinsic expression of representative antiviral ISGs leads to susceptibility of hTSCs to ZIKV

Intrinsically expressed ISGs serve as cell-autonomous antiviral effectors for host antiviral defense[34]. To explore the intrinsic expression of antiviral ISGs in different trophoblast cell types, we performed RNA-seq on hTSCs, STB[TS] and EVT[TS]. We used the expression level of *HLA-G* as a criterion to evaluate the intrinsic expression level of ISGs in hTSC-derived trophoblast cells. We found an overlapping set of intrinsically highly expressed ISGs in hTSCs and hTSC-derived STB[TS] and EVT[TS], including *IF16, ODC1, LY6E, SAT1, CDKN1A, EIF3L, MCL1, DDX3X, ISG15, IFNGR1* and *TNFAIP3*, which were previously reported to be associated with antiviral functions (Fig. 3a and b)[35-44]. However, all hTSC-derived trophoblast cell types lacked the expression of many representative antiviral ISGs, such as *IRFs, APOLs, TLR3/7, CCLs, ISG20, IFNLR1, BST2* and *APOBEC3G* (Fig. 3b). In particular, the expression of *IFIT* family and *IFITM* family genes, which have broad-spectrum antiviral functions, was low in all types of hTSC-derived trophoblast cells (Fig. 3b). The intrinsic expression of the ISGs were also validated by qRT-PCR, which

showed that *IFITs, IFITMs, TLRs* and *IRFs* were indeed expressed at low levels in hTSC-derived trophoblast cells (Fig. 3c).

We next analyzed the specific expression pattern of ISGs among hTSCs, STB[TS] and EVT[TS]. We found several ISGs highly expressed in hTSCs, which were decreased after differentiation, such as *PNPT1, PABPC4* and *CREB3L3* (Fig. 3b and S5A). Interestingly, the ISGs specifically expressed in hTSCs had no known direct antiviral functions. In contrast, some ISGs involved in the innate immune responses following ZIKV infection and in the inhibition of RNA virus replication were elevated with hTSC differentiation, such as *NFIL3, ATF3* and *RIPK2*, which may be associated with increased resistance to ZIKV infection during differentiation (Fig. 3b and S5B)[45-47]. Of note, no specific intrinsic ISGs were observed in STB[TS] (Fig. 3a). *NOS2*, which was reported to play a major role in host protection in flavivirus infection, was highly expressed in EVT[TS] [48]. In addition, functional isoforms of *MT1*, including *MT1G, MT1H, MT1F* and *MT1X*, were highly expressed in EVT[TS] (Fig. 3b and S5C). The results of RNA-seq analysis were validated by qRT-PCR (Figure S5). To verify the generalizability of the ISG expression profile in hTSCs of different origins and primary trophoblast cells, we analyzed ISG expression in the scRNA-seq data from primary placental tissues and trophoblast organoids built with other hTSC lines published by Shannon et al.[49]. We found that the expression profile of ISGs was conserved in hTSCs of different origins and in primary trophoblast cells, and trophoblast cells were indeed low expressed representative antiviral ISGs (Figure S6).

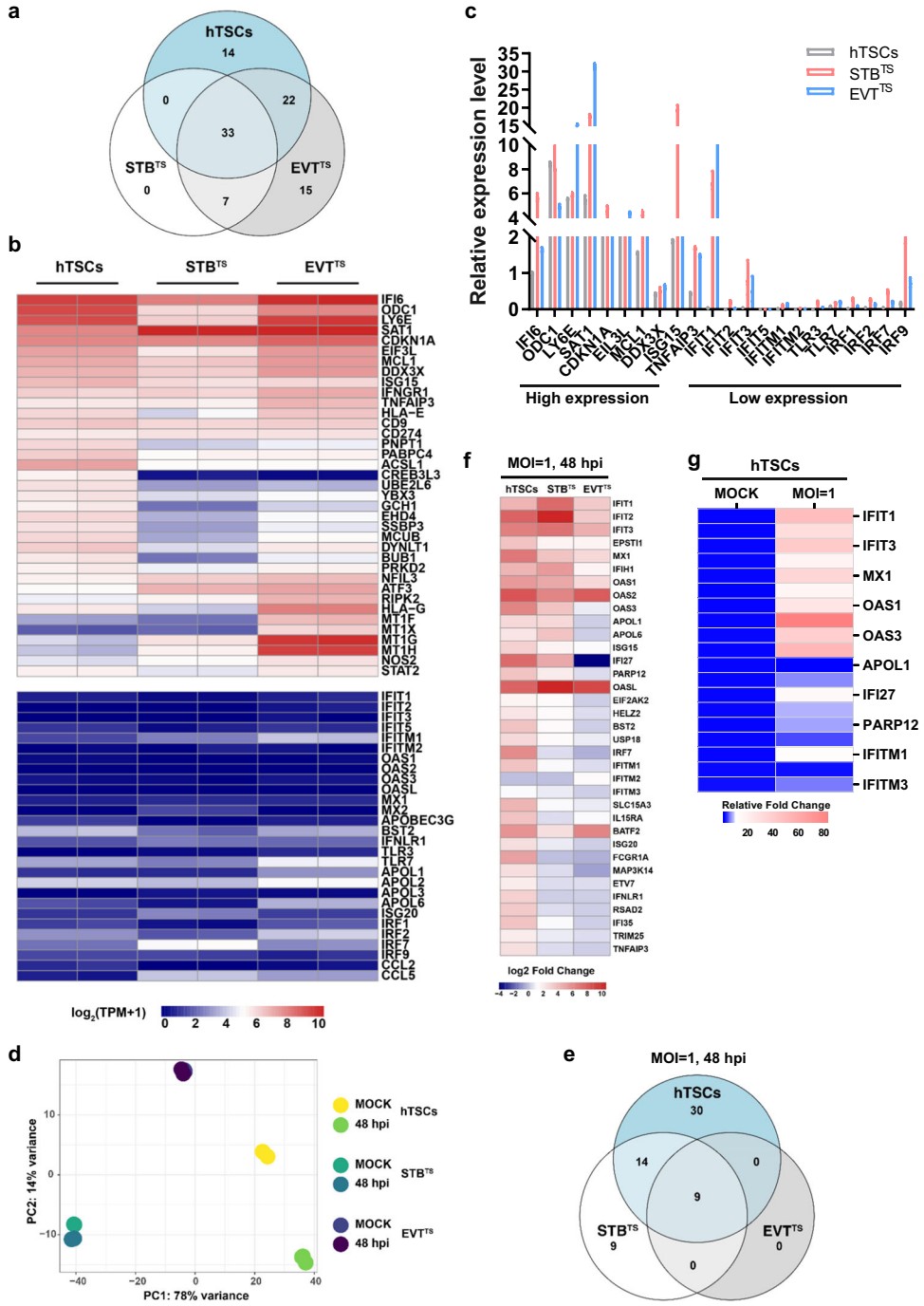

**Fig. 3 | hTSCs lack intrinsic expression of representative antiviral ISGs. a** Venn diagram showing the number of ISGs specifically expressed in hTSCs, STB^TS and EVT^TS, and shared by all the cell types. **b** Heatmap showing the z-score TPM of ISGs with high and low expression in hTSCs, STB^TS and EVT^TS. **c** Quantification of the expression of ISGs with high and low expression shown by RNA-seq in hTSCs, STB^TS and EVT^TS by qRT-PCR. *n* = 3 independent experiments. Data are shown as the mean ± s.d. **d** PCA analysis of the transcriptomes of mock- and ZIKV-infected hTSCs, STB^TS and EVT^TS. The cells were exposed to ZIKV at an MOI of 1, and analyzed at

48 hours post infection. **e** Venn diagram showing the number of ISGs specifically up-regulated in hTSCs, STB^TS and EVT^TS following ZIKV infection and shared by all the cell types. **f** Heatmap showing the up-regulated ISGs in hTSCs, STB^TS and EVT^TS. **g** Quantification of the relative expression of ISGs in ZIKV-infected hTSCs compared to in mock-infected hTSCs by qRT-PCR. The hTSCs were infected with ZIKV at an MOI of 1 and the infected cells were separated by FACS for qRT-PCR analysis at 48 hours post infection. *n* = 3 independent experiments.

Then, we examined the ISG responses to ZIKV infection in hTSCs, STB^TS and EVT^TS. Considering the low proportion of infected cells when infected with ZIKV at an MOI of 0.1, we infected the hTSCs, STB^TS and EVT^TS with ZIKV at an MOI of 1, and both the growth of viral RNA in the supernatants and the proportion of infected cells increased compared to those in the cells infected with ZIKV at an MOI of 0.1 (Figure S7). Therefore, we performed RNA-seq analysis on cells infected with ZIKV

at an MOI of 1. Principal Component Analysis (PCA) revealed that hTSCs exhibited dramatic transcriptome variations, while minimal changes were observed in both STB^TS and EVT^TS after infection (Fig. 3d), which was attributed to the low infection rate of ZIKV in STB^TS and EVT^TS. We analyzed the up-regulated ISGs with a fold change>4 in hTSCs and a fold change>2 in STB^TS and EVT^TS following ZIKV infection. We found a few overlapping ISGs between hTSCs and STB^TS and EVT^TS

(Fig. 3e). ISGs with well-known antiviral activity, including *IFITs*, *IFIH1*, *MX1* and *EPSTI1*, were elevated in all types of trophoblast cells (Fig. 3f). ZIKV-infected hTSCs and STB[TS] shared some known anti-flavivirus ISGs, such as *OASs*, *OASL*, *APOL1/6*, *ISG15*, *IFI27*, *PARP12*, *EIF2AK2*, *HEL2*, *BST2* and *USP18* (Fig. 3f). Moreover, a number of ISGs associated with the antiviral innate immune response were specifically up-regulated in hTSCs, such as *SLC15A3*, *IL15RA*, *BATF2*, *ISG20*, *FCGR1A*, *MAP3K14*, *IRF7*, *ETV7*, *IFNLR1*, *RSAD2*, *IFI35*, *TRIM25* and *TNFAIP3*. Interestingly, *IFITM1* and *IFITM3*, which are restriction factors blocking the entry of many viruses, were increased in hTSCs after ZIKV infection (Fig. 3f). The excessive expression of *IFITMs* is known to impair STB formation, suggesting that the syncytialization of ZIKV-infected hTSCs might be restricted[50].

To further validate the different infectivity of ZIKV to hTSCs, STB[TS] and EVT[TS], we separated infected and uninfected cells by flow cytometry (FACS) using ZIKV E protein antibody, which showed a lower percentage of infected cells in STB[TS] and EVT[TS] than in hTSCs (Fig. S8). Since hTSCs served as the main target cells for ZIKV infection, we sorted ZIKV-infected hTSCs by FACS and validated the changed ISGs after ZIKV infection by qRT-PCR analysis, indicating that several important antiviral ISGs were activated following ZIKV infection, including *IFITs*, *IFIH1*, *EPSTI1*, *OASs*, *OADL*, *APOL6*, *IFI27*, *ISG15*, *PARP12* and *IFITM1/3*, which was consistent with the results of RNA-seq (Fig. 3g).

We wondered whether the low intrinsic expression of antiviral ISGs resulted in low expression of IFNs in trophoblast cells as well, thus we profiled intrinsic IFN expression in hTSCs, STB[TS] and EVT[TS], including *IFNα*, *IFNβ*, *IFNγ*, *IFNδ*, *IFNε*, *IFNλ1*, *IFNλ3*, *IFNτ*, *IFNω*. We found that only few IFNs were intrinsically expressed in trophoblast cells, and no representative antiviral IFNs were intrinsically expressed in trophoblast cells (Fig. S9A). Then, we analyzed the IFN expression in the hTSCs, STB[TS] and EVT[TS] infected with ZIKV at an MOI of 1. We found that apart from the genes involved in ISGs, some essential antiviral IFNs, such as *IFNL1* and *IFNL3*, were activated in all types of trophoblast cells (Fig. S9B). These results showed the activation of innate immune responses in ZIKV-infected hTSCs, which may disrupt the development of trophoblast cells and the structure of the developing placenta during early embryo development[51]. Taken together, the results showed that hTSCs lacked intrinsic expression of representative antiviral ISGs and IFNs. Upon infection, hTSCs exhibited extensive activation of antiviral innate immune responses.

## Modeling exposure and infection to ZIKV disrupts the structure of hTSC-organoids

Placental villous trophoblast cells function as an effective barrier to prevent vertical transmission of maternal pathogens during pregnancy. To further investigate the characteristics and consequences of ZIKV infection on human placental trophoblasts, we established a placental trophoblast organoid and subjected it to ZIKV infection. We seeded hTSCs into Matrigel drops and allowed them to develop for 6 days in vitro (Fig. 4a). The size of the aggregate grew with culturing and could be maintained by passaging (Figs. 4a and S10A). After 6 days of culture, IF staining confirmed that the cells in the periphery were proliferative mononucleated CTB expressing Ki67 and CDH1, and the mononucleated cells fused in the center expressing CGB, which resembled the STB within the placental villi (Fig. 4b). The structure was defined as hTSC-organoid. The hTSC-organoids showed higher expression of the trophoblast markers *GCM1*, *ERVW-1* (*SYNCYTIN-1*), *ELF5* and chromosome 19 microRNA cluster (C19MC) than the positive control JEG-3[24,27,52,53] (Figure S10B and S10C). The hTSC-organoids could also secrete placental hormones, such as Human Chorionic Gonadotropin β (hCG-β) and GDF15, and could generate EVT with induction of NRG-1, A83-01 (an inhibitor of TGF-β), and Matrigel[26,54–56] (Fig. S10D–S10F). These results showed that the hTSC-organoids could recapitulate the secretory activity, molecular signature and differentiation ability of original human placental trophoblast cells.

Using the hTSC-organoids, we evaluated the permissiveness and infection characteristics of hTSC-organoids to ZIKV. Mature hTSC-organoids (Day 6) were exposed to ZIKV at an MOI of 10 for 12 hours and the viral RNA in hTSC-organoids and supernatants was detected at 6, 12, 24 and 48 hpi by qPCR. The growth of viral RNA was observed in both organoids and supernatants, suggesting that the hTSC-organoids supported ZIKV infection and replication (Fig. 4c). We further characterized the hTSC-organoids infected with ZIKV at MOIs of 1 and 10 by IF staining. As shown in Fig. 4d, at an MOI of 1, ZIKV E protein signals were mainly observed in the CTB of hTSC-organoids. When the virus exposure was increased to an MOI of 10, intracellular E protein signals were observed in both CTB and STB (Fig. 4d).

Through an in vitro hTSC-organoid-based ZIKV infection model, we evaluated the effect of ZIKV infection on the structural development of mature hTSC-organoids. The mature hTSC-organoids (Day 6) were inoculated with ZIKV at an MOI of 10 and were detected at 3 and 5 days after infection (Fig. 4e). We found that the hTSC-organoids were significantly broken up after ZIKV infection compared to the mock-infected hTSC-organoids (Fig. 4f). Taken together, these results demonstrated that hTSC-organoid was a physiological model of human placental trophoblast, supporting ZIKV infection and replication. Through an hTSC-organoid-based ZIKV infection model, we observed that ZIKV infection disrupted the hTSC-organoid structure.

## Single-cell RNA sequencing reveals the cellular complexity of hTSC-organoids

To characterize ZIKV infection of hTSC-organoids, Day 8 mock-infected and ZIKV-infected (MOI 1 and 10) hTSC-organoids were digested into single cells for scRNA-seq using the 10x Genomics platform. After strict exclusion of cells with low quality, high expression of mitochondrial genes and no expression of trophoblast cell markers, 28617, 17854 and 17230 high-quality cells from mock- and ZIKV (MOI 1 and 10)-infected hTSC-organoids were used for subsequent analysis. Uniform manifold approximation and projection (UMAP) analysis showed that hTSC-organoids comprised eleven (Clusters 0-10) discrete trophoblast subpopulations (Figs. 5a, S11A−S11C). To determine the cell types within hTSC-organoids, we evaluated the expression of trophoblast marker genes across distinct subpopulations (Fig. 5b). The pan-trophoblast markers *KRT8* and *KRT18* were strongly expressed in all the cells. Cluster 1 cells were enriched in the expression of the cell cycle-related genes *RRM2*, *CDK1* and *CCNB1*, which were considered proliferative mature CTB, defined as 1_CTB_1 (Figs. 5b and S11E). Cluster 2 cells were enriched in the expression of the genes regulating CTB fusion to STB, including *GCM1*, *OVOL1*, *ERVW-1* and *ERVFRD-1*, and were defined as 2_CTB_Fusion (Figs. 5b and S11D). Cluster 8 cells were enriched in the expression of STB marker genes, including *CGA*, *CGB3*, *CGB5*, *CGB7*, *CGB8* and *SDC1*, which were considered mature STB, and were defined as 8_STB_Mature (Figs. 5b and S11F). The hTSC-organoids established in this study showed a similar expression profile of trophoblast marker genes with those in the trophoblast organoids constructed with other hTSC lines, indicating that the hTSCs used in this study are representative (Figure S12).

To investigate the developmental trajectory of trophoblast cells within the hTSC-organoids, we performed RNA velocity analysis. The 1_CTB_1 cells were differentiated from Cluster 0 and 6 cells (Fig. 5c), thus, Cluster 0 and 6 cells were considered hTSCs, and were defined as 0_hTSC_1 and 6_hTSC_2. The cells in 1_CTB_1 were differentiated into 2_CTB_Fusion through Clusters 3, 4 and 5. Therefore, Cluster 3, 4 and 5 cells were considered developing CTB, and were defined as 3_CTB_2, 4_CTB_3 and 5_CTB_4, respectively. The CTB marker genes *TEAD4* and *CDH1* were highly expressed in the cells belonging to CTB (1_CTB_1, 3_CTB_2, 4_CTB_3 and 5_CTB_4). The cells in Clusters 7, 9 and 10 were differentiated from 2_CTB_Fusion, and eventually differentiated into

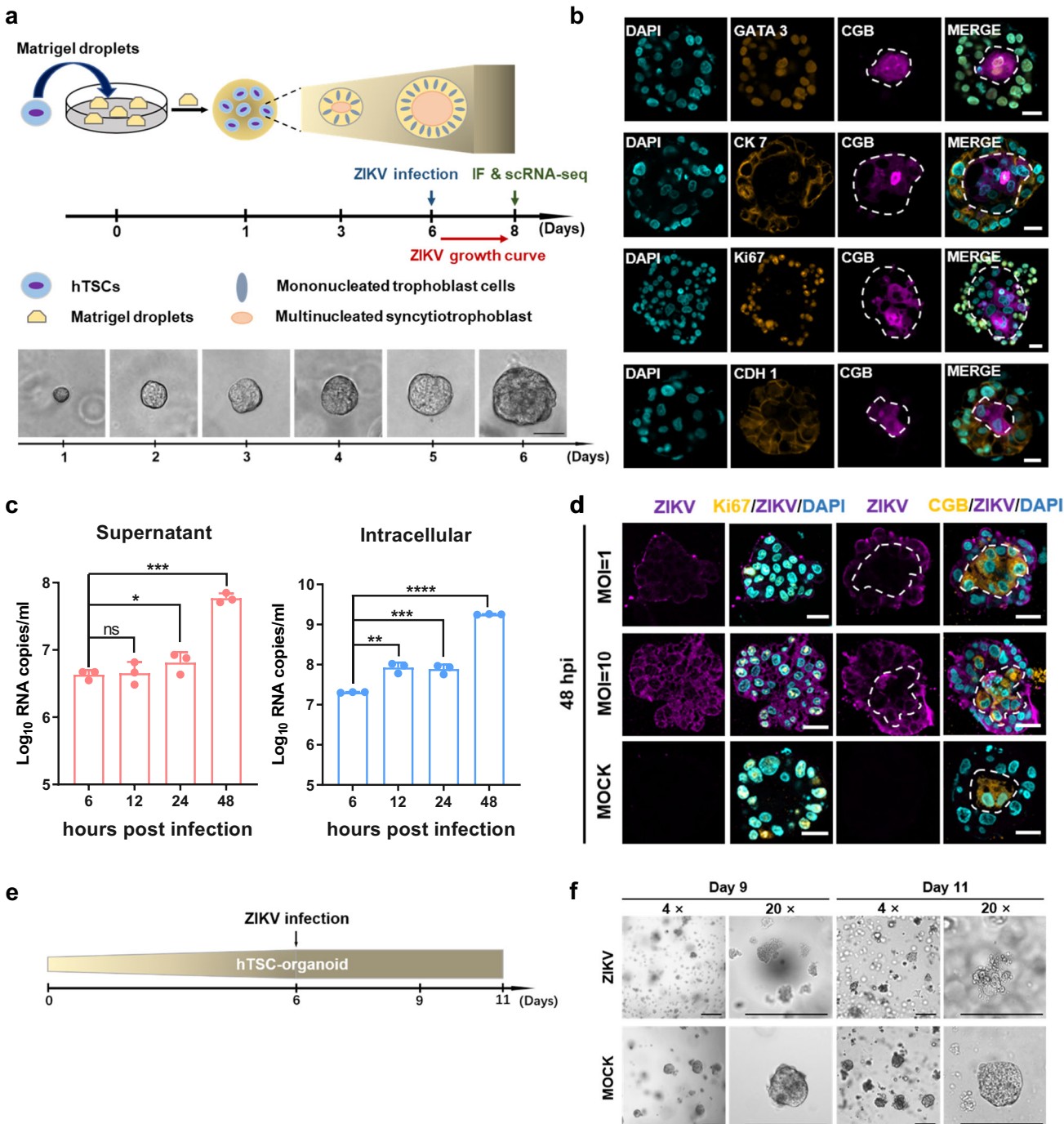

**Fig. 4 | Modeling exposure to and infection with ZIKV disrupts the structure of hTSC-organoids. a** A schematic diagram of the construction of hTSC-organoids and establishment of ZIKV infection, and brightfield images of hTSC-organoid growth. Scale bar: 50 μm. **b** Immunofluorescence staining analysis for the cell types in hTSC-organoids. Pan-trophoblast cell markers, GATA 3 and CK 7. Proliferative CTB markers, Ki67 and CDH 1. STB marker, CGB. Nuclei were stained with DAPI. The white dished line showing the outline of the multinucleated STB in hTSC-organoids. Scale bars: 20 μm. **c** Quantification of ZIKV RNA in the supernatants (left panel) and intracellular (right panel) of hTSC-organoids at 6, 12, 24 and 48 hours post infection. Two-tailed unpaired t test was used for statistical analysis of significance. *n* = 3 independent experiments. Supernatant, *, *p* = 0.035. ***, *p* = 0.0007. ns, no significance. Intracellular, **, *p* = 0.0011. ***, *p* = 0.0009. ****, *p* < 0.0001. Data are shown as the mean ± s.d. **d** Immunofluorescence staining of mock- and ZIKV-infected hTSC-organoids for ZIKV E protein, Ki67 and CGB. The representative images were selected from three independent experiments. Nuclei were stained with DAPI. The hTSC-organoids were infected with ZIKV at MOIs of 1 and 10, and analyzed at 48 hours post infection. The white dished line showing the outline of the multinucleated STB in hTSC-organoids. Scale bars: 20 μm. **e** A schematic diagram of ZIKV infection of mature hTSC-organoids. Day 6 hTSC-organoids were exposed to ZIKV at an MOI of 10, and were analyzed at day 9 and day 11. **f** Representative brightfield images showing the structure of mock- and ZIKV-infected hTSC-organoids at day 9 and day 11, which were selected from three independent experiments. Scale bars, 50 μm.

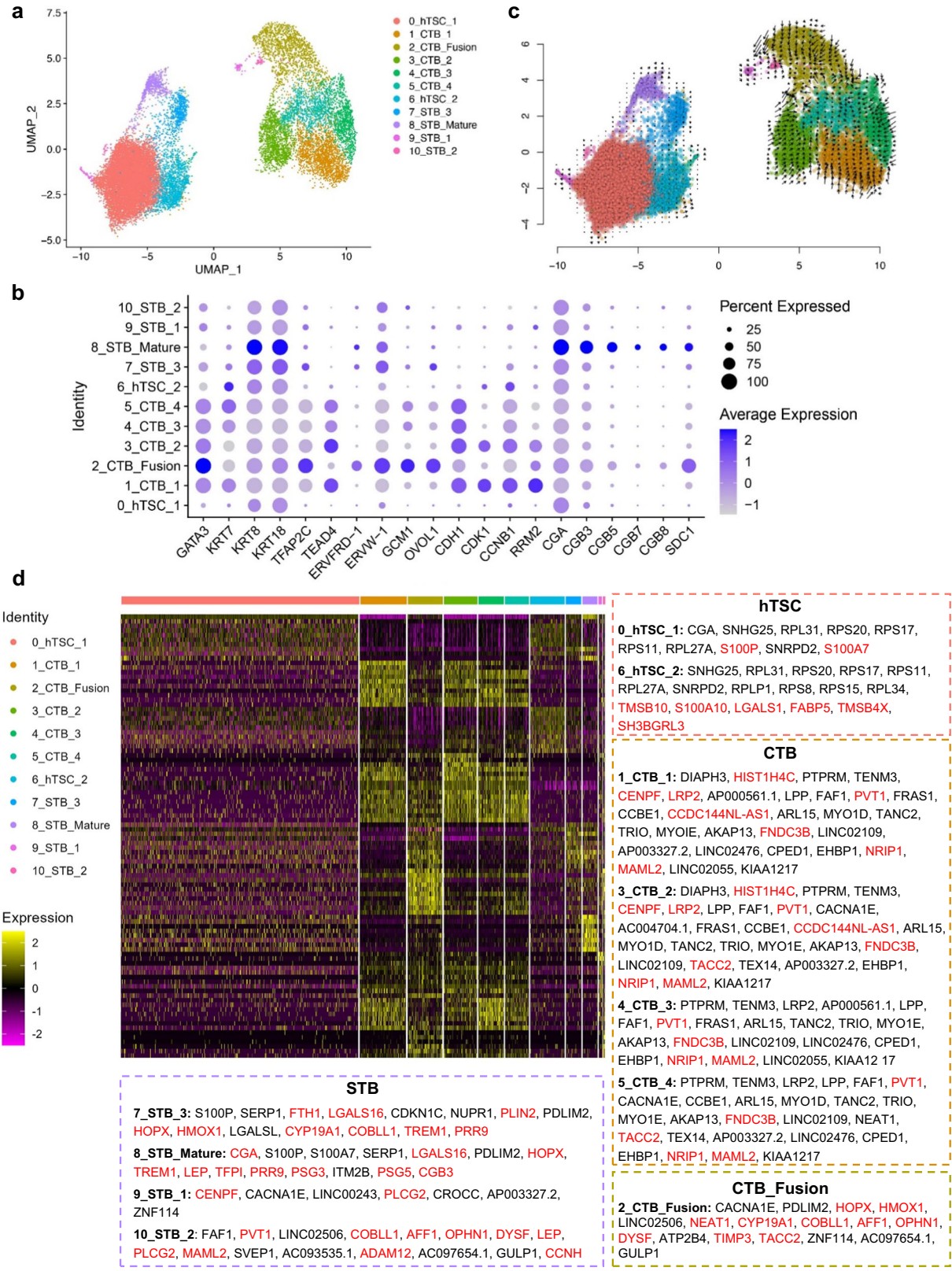

**Fig. 5 | Single-cell transcriptome profiles of hTSC-organoids. a** UMAP showing the cell composition of hTSC-organoids. **b** Dot plot indicating the expression of trophoblast markers in distinct clusters of hTSC-organoids. The percentage of cells that express each gene and their average gene expression levels were presented with differential circle sizes and color intensities, respectively. **c** UMAP overlaid with the RNA velocity of hTSC-organoids. Black arrows represented the calculated velocity trajectories. **d** Heatmap showing the top 10 differentially expressed genes for different cell clusters of hTSC-organoids. The genes indicated in the main text were highlighted in red.

8_STB_Mature. Therefore, Cluster 7, 9 and 10 cells were considered developing STB, and were defined as 7_STB_3, 9_STB_1 and 10_STB_2, respectively.

To further characterize the trophoblast subpopulations in hTSC-organoids, we analyzed the top 10 differentially expressed genes (DEGs) between different clusters (Fig. 5d). In the cells belonging to hTSCs (0_hTSC_1 and 6_hTSC_2), the genes associated with stemness of trophoblast progenitor cells, including S100 family genes (*S100P*, *S100A7* and *S100A10*), *TMSB4X* and *SH3BGRL3*, and trophoblast genes, including *TMSB10*, *LGALS1* and *FABP5*, were highly expressed. The cells belonging to CTB (1_CTB_1, 3_CTB_2, 4_CTB_3 and 5_CTB_4) were enriched for trophoblast proliferation-related genes *HISTH4C*, *CENPF* and *PVT1* and trophoblast genes *LRP2*, *CCDC144NL-AS1*, *FNDC3B*, *NRIP1* and *MAML2*. With the differentiation of CTB, the expression of the proliferation-related genes *HIST1H4C* and *CENPT* decreased in 4_CTB_3 and 5_CTB_4 and the fusion-related gene *TACC2* increased in 5_CTB_4. The cells in 2_CTB_Fusion highly expressed the genes regulating the fusion and proliferation of trophoblast cells, including *HOPX*, *HMOX1*, *NEAT1*, *CYP19A1*, *COBLL1*, *AFF1*, *OPHN1*, *DYSF*, *TIMP3* and *TACC2*. Among the clusters belonging to STB, Clusters 9_STB_1 and 10_STB_2 had few cells and retained the expression of some genes regulating cell proliferation, including *CENPF*, *PVT1*, *COBLL1*, *AFF1*, *OPHN1* and *PLGG2* and trophoblast genes, including *DYSF*, *LEP*, *MAML2*, *ADAM12* and *CCNH*. The cells belonging to 7_STB_3 and 8_STB_Mature highly expressed STB marker genes, including *CYP19A1*, *CGA*, *LGALS16*, *HOPX*, *HMOX1*, *TREM1*, *TFPI*, *PRR9*, *PSG3*, *PSG5* and *CGB3* and the trophoblast genes *FTH1* and *PLIN2*. Taken together, the trophoblast cells in hTSC-organoids simulated the syncytialization of trophoblast cells from CTB to CTB_Fusion to STB.

### Characterization of ZIKV-infected hTSC-organoids at the single-cell level

To explore the effect of ZIKV infection on different trophoblast cell types in hTSC-organoids, we grouped the cells in hTSC-organoids into 4 major cell subtypes, including hTSC (0_hTSC_1 and 6_hTSC_2), CTB (1_CTB_1, 3_CTB_2, 4_CTB_3 and 5_CTB_4), CTB_Fusion (2_CTB_Fusion) and STB (7_STB_3 and 8_STB_Mature) (Fig. 6a). Although the cells in 9_STB_1 and 10_STB_2 were developed from 2_CTB_Fusion, the STB marker genes were expressed at lower levels than those in the cells in 7_STB_3 and 8_STB_Mature, therefore, they were considered developing STB and were not grouped as STB. To distinguish the infected cells at the single-cell level, *MX1*-positive cells were considered ZIKV-infected trophoblast cells since *MX1* is known to be increased after ZIKV infection, which was also demonstrated in ZIKV-infected hTSCs, STBᵀˢ and EVTᵀˢ (Fig. 3f and g). *MX1*-positive cells were used for subsequent comparative analysis with mock-infected cells.

We compared the proportions of ZIKV-infected CTB, CTB_Fusion and STB in the hTSC-organoids infected with ZIKV at MOIs of 1 and 10. A higher proportion of infected cells was found in the hTSC-organoids infected with ZIKV at an MOI of 10 than in the hTSC-organoids infected with ZIKV at an MOI of 1 (Figs. 6b, S13A and S13B). ZIKV-infected hTSC-organoids also showed decreased susceptibility to ZIKV with the syncytialization of CTB (Fig. 6b). The ISGs and IFNs showed similar changes in the *MX1*-positive hTSCs of ZIKV-infected hTSC-organoids to those in ZIKV-infected 2D cultured hTSCs (Figs. 6b and S13C). The marker genes of trophoblast progenitor cells, including *TEAD4*, *TP63*, *CDH1* and *ITGA6*, were decreased in the *MX1*-postive hTSCs of ZIKV-infected hTSC-organoids, suggesting a disruption of hTSC stemness (Fig. 6d). The STB was considered to be the main source of polypeptide hormones important for pregnancy maintenance. We evaluated the changes in polypeptide hormone-encoding genes in the *MX1*-positive STB from ZIKV-infected hTSC-organoids, which showed similar changes to those in the maternal circulation or placenta from pregnant women with PE, including increased *CGA*, *CGB* family genes, *CRH*, *GIP*, *INHA*, *INSL4*, *LEP*, *LHB* and

*TAC3* and decreased *ADM*, *GAL*, *ACTN1* and *ACTN4*, suggesting a possible cause of PE by ZIKV infection (Fig. 6e)[57].

To explore the effect of ZIKV infection on CTB and CTB_Fusion, we analyzed the DEGs (Log₂ fold changes>0.1 and Log₂ fold changes < −0.1) in ZIKV-infected CTB and CTB_Fusion. In ZIKV-infected CTB, the up-regulated genes were enriched in virus infection-related signaling pathways, while the down-regulated genes were enriched in cell cycle-related signaling pathways, suggesting an inhibition of CTB proliferation by ZIKV infection (Fig. 6f and g). In ZIKV-infected CTB_Fusion cells, the genes involved in GnRH signaling pathway, fluid shear stress and atherosclerosis and thyroid hormone synthesis were up-regulated (Fig. 6h and i). The genes belonging to the Hippo signaling pathway, which is essential for the self-renewal of trophoblast progenitor cells, were decreased in CTB_Fusion cells after ZIKV infection (Fig. 6h and i). Taken together, using scRNA-seq, we found that ZIKV infection may disrupt the stemness of hTSCs and the proliferation of CTBs, and lead to a PE phenotype.

### ZIKV infection represses the syncytialization of hTSC-organoids

To explore the long-term effects of ZIKV infection on the development of placental villi, we designed a development assay using hTSC-organoids. We exposed the day 1 hTSC-organoids to ZIKV at an MOI of 10 and visualized the development of the infected-hTSC-organoids on day 8 (Fig. 7a). We observed a restricted development of STB and a disordered distribution of STB and CTB in ZIKV-infected hTSC-organoids, suggesting the adverse effects of ZIKV infection on trophoblast development (Fig. 7b). Insufficient syncytialization has been well-known to result in pathological conditions such as PE[58]. In contrast, mock-infected hTSC-organoids developed normally (Fig. 7b).

To further verify the inhibition of syncytialization by ZIKV infection, we subjected hTSCs to ZIKV at an MOI of 1 and then induced STB differentiation for 96 hours (Fig. 7c). With differentiation in STB medium, we found that the expression of *CGB* and *SDC1* (markers of STB) in the STBᵀˢ differentiated from the ZIKV-infected hTSCs was lower than that from mock-infected hTSCs by qRT-PCR (Fig. 7d). We further performed IF staining for CGB and CDH1 in the STBᵀˢ differentiated from mock- and ZIKV-infected hTSCs, revealing reduced CGB protein after ZIKV infection (Fig. 7e). We confirmed that the fusion capacity of hTSCs was significantly decreased after ZIKV infection compared to that of mock-infected hTSCs by calculating the fusion index (Fig. 7f, Table S3 and S4). IF staining for ZIKV E protein and CDH1 in the STBᵀˢ differentiated from ZIKV-infected hTSCs for 96 hours showed that almost none of the infected hTSCs underwent cell fusion (Figure S14).

During syncytialization of hTSCs, cAMP-PKA-CREB-STAT5B signaling induced trophoblast cell fusion through the upregulation of fusogenic *ERVW-1* (*SYNCYTIN 1*) and *ERVFRD-1* (*SYNCYTIN 2*) by the transcription factors *GCM1* and *OVOL1*, and resulted in CGB production (Fig. 7g)[59–61]. We observed a down-regulated expression of *GCM1*, *OVOL1*, *ERVW-1*, and *ERVFRD-1* in hTSCs following ZIKV infection (Fig. 7h). Taken together, these results demonstrated that ZIKV infection suppressed the syncytialization of hTSC-organoids and resulted in decreased expression of fusogenic genes within the cAMP pathway in hTSCs.

### Discussion

In this study, to investigate the infectivity and pathogenicity of ZIKV on trophoblast cells, we established ZIKV infection on hTSC-derived trophoblast cells and organoids. We demonstrated that hTSCs were the main target cells of ZIKV. *AXL* and *TIM-1* played important roles in ZIKV infection of trophoblast cells. ZIKV infection disrupted the structure of hTSC-organoids and inhibited syncytialization. By isolating ZIKV-infected 2D cultured trophoblast cells using FACS and identifying ZIKV-infected cells in hTSC-organoids using scRNA-seq for transcriptome analysis, we demonstrated that hTSCs lack intrinsic

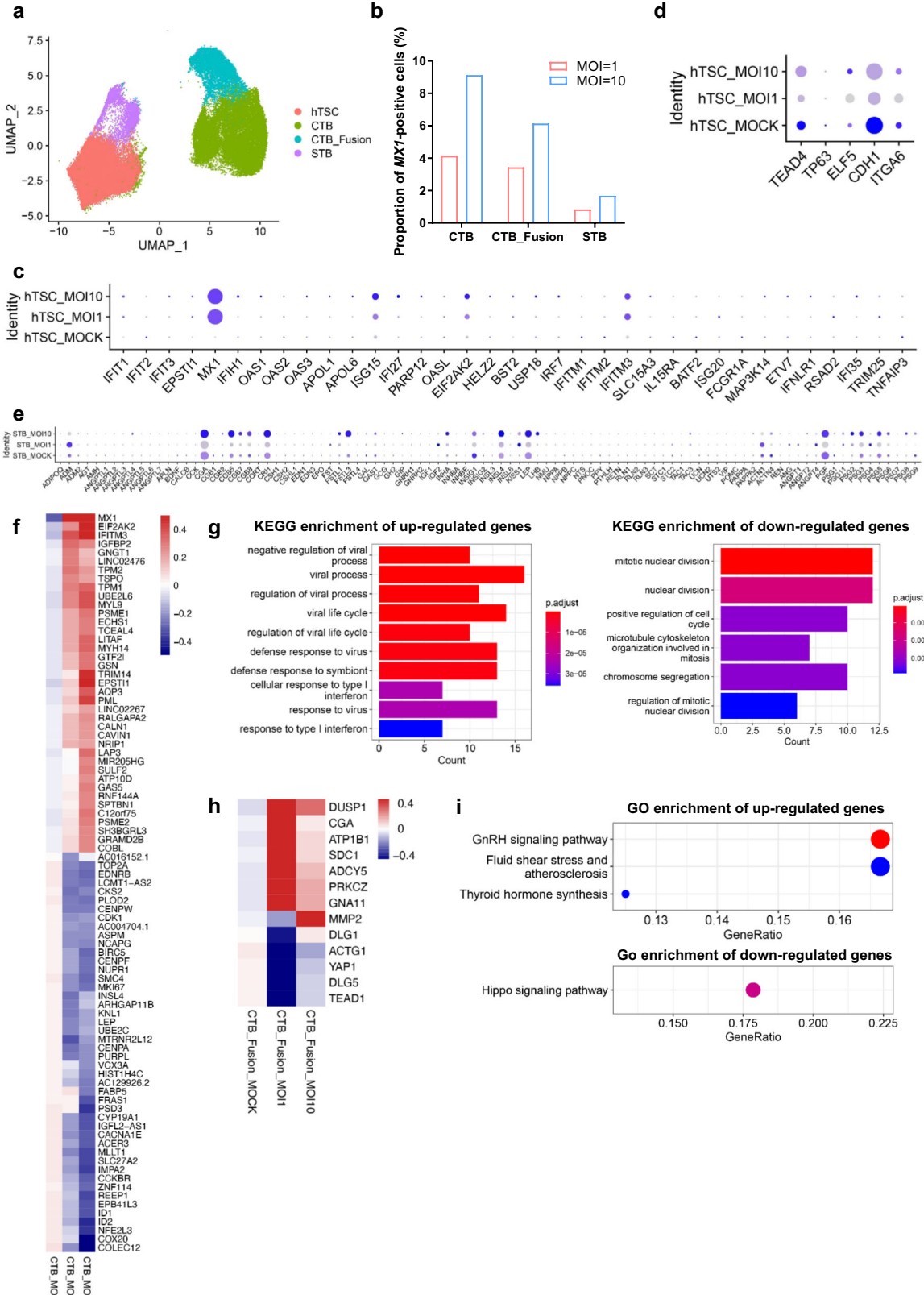

expression of representative antiviral ISGs and IFNs. The results of scRNA-seq showed that ZIKV infection may disrupt hTSC stemness and CTB proliferation and may result in a PE phenotype.

Intrinsic expression of ISGs is essential for host cell resistance to viral infection. For example, IFIT and IFITM proteins showed a broad-spectrum of antiviral activities against viral infection, and placental cells protected trophoblast cells from ZIKV infection through auto-crine or paracrine of type III IFN[12,62]. Wu et al. showed that hESCs were resistant to a diverse panel of viruses, which was due to the high baseline expression of representative ISGs, such as *IFITMs*, *BST2* and *EIF3L*. In contrast, here we showed that hTSCs lacked the expression of representative antiviral ISGs and IFNs. Differences in the intrinsic

**Fig. 6 | Characterization of ZIKV-infected hTSC-organoids at the single-cell level. a** UMAP showing the cell composition of mock- and ZIKV (MOI 1 and 10)-infected hTSC-organoids. The cells from mock- and ZIKV-infected hTSC-organoids were integrated and grouped into four cell types, namely, hTSC, CTB, CTB_Fusion and STB. **b** Proportion of MX1-positive cells in the CTB, CTB_Fusion and STB groups from hTSC-organoids infected with ZIKV at MOIs of 1 and 10. **c** Dot plot indicating the expression of ISGs in the hTSCs from mock- and ZIKV (MOI 1 and 10)-infected hTSC-organoids. **d** Dot plot indicating the expression of the marker genes of trophoblast progenitor cells in the hTSCs from mock- and ZIKV (MOI 1 and 10)-infected hTSC-organoids. **e** Dot plot indicating the expression of the polypeptide hormone-encoding genes in the STB from mock- and ZIKV (MOI 1 and 10)-infected hTSC-

organoids. **f** Heatmap showing the differentially expressed genes (Log$_2$ fold changes>0.1 and Log$_2$ fold changes < −0.1) in the CTB from mock- and ZIKV (MOI 1 and 10)-infected hTSC-organoids. **g** KEGG enrichment of the signaling pathways for up- (left panel) and down (right panel)-regulated genes in ZIKV-infected CTB. The $p$ value was adjusted using Benjamini & Hochberg method with a cutoff of $p$ value = 0.01 and $q$ value = 0.05 on the enrichment tests. **h** Heatmap showing the differentially expressed genes in CTB_Fusion from mock- and ZIKV (MOI 1 and 10)-infected hTSC-organoids. **i** GO enrichment of the signaling pathways for up- (upper panel) and down (lower panel)-regulated genes in CTB_Fusion. The number of the genes and $p$ value of each gene are presented with differential circle sizes and color intensities, respectively.

expression of antiviral ISGs and IFNs may result in distinct resistance to viral infection. After ZIKV infection, essential antiviral ISGs, such as *IFITs* and *IFITMs*, and essential antiviral IFNs, such as *IFNL1* and *IFNL3*, were greatly activated in hTSCs. The activation of IFN signaling after infection may result in the disruption of placental morphology and may mediate pregnancy complications[51]. In this study, the host immune response of trophoblast cells to ZIKV infection was obtained using bulk RNA-seq and validated using qRT-PCR after sorting out infected and uninfected cells by FACS, which accuracy will be further improved by recently developed virus-inclusive scRNA-seq techniques[63–66].

What are appropriate in vitro models for studying the tissue tropism of viruses? Miniature organoids have been shown to simulate the development, structure and function of organs. A variety of in vitro organoids stimulating tissues have been used in viral studies, such as lung organoids, liver organoids, kidney organoids, brain organoids and gut organoids[67]. Organoid-based viral infection models have greatly advanced the understanding of viral damage to the body, such as neuronal infections with ZIKV, intestinal infections with enteroviruses and respiratory infections with SARS-CoV-2[67,68]. Recently, Theunissen et al. described the use of trophoblast organoids to probe placental susceptibility SARS-CoV-2 and ZIKV[28]. Here, we developed hTSC-based human trophoblast organoids, which can simulate the syncytialization of trophoblast cells. Using an hTSC-organoid-based ZIKV infection model, we showed that ZIKV infection disrupted the structural development and syncytialization of hTSC-organoids. We demonstrated the possibility of disrupting the stemness of hTSCs and the proliferation of CTBs and causing a PE phenotype by ZIKV infection at the single-cell resolution. Compared to existing cellular models, our hTSC-organoid-based infection model can better represent the damage of ZIKV infection on human placenta. In conclusion, the in vitro organoid system was very helpful in increasing the understanding of tissue tropism toward viral infection.

The reasons for the different sensitivity of different trophoblast cell types to ZIKV remain largely unknown, owing to the lack of appropriate and flexible cellular models[11,12,22]. The lack of models for simulating early placental development also leads to unknown infectivity of ZIKV in placental progenitor cells during early human embryonic development. Sheridan et al. have shown the vulnerability of human trophoblast cells to ZIKV and evaluated the expression of potential host factors for flavivirus in human trophoblast cells using hESC-derived trophoblast-like cells. The hESC-derived trophoblast-like cells showed a high expression of *TAM* family genes (*AXL*, *TYRO3* and *MERTK*), and low expression of the *TIM* family gene (*TIM-1*)[16]. However, some differences were indicated between the primary hTSCs and hESC-derived trophoblast-like cells, including the differential expression of *TIM-1*, and we confirmed that hESC-derived trophoblast-like cells (H1-TS) had much lower expression of *TIM-1* than blastocyst-derived hTSCs (Figure S15)[11,12,20,24]. Here, using blastocyst-derived hTSCs, we found decreased susceptibility to ZIKV after the differentiation of hTSCs. We demonstrated the important roles of *AXL* and *TIM-1* in ZIKV infection of trophoblast cells by gene knockout and

overexpression of *AXL* or *TIM-1* in EVT$^{TS}$ enhanced susceptibility to ZIKV infection. The role of *AXL* and *TIM-1* in ZIKV infection of human host cells was also studied in other tissues. Nowakowski et al. highlighted the high expression of *AXL* in human neural stem cells and a set of diverse neural cell types[33]. Chen et al. demonstrated that *AXL* promoted ZIKV infection in astrocytes by antagonizing type I IFN signaling[15]. However, Wells et al. showed that genetic ablation of *AXL* could not protect neural progenitor cells from ZIKV infection[31]. In conclusion, the role of *AXL* and *TIM-1* differs in ZIKV infection of different cell types.

In summary, our study demonstrated that hTSCs provided physiological, flexible and reliable models for studying the infectivity and infection mechanism of ZIKV on placental trophoblast cells. The self-organized trophoblast organoids established using hTSCs can simulate the syncytialization of trophoblast cells. The 3D hTSC-organoid-based ZIKV infection model could better simulate the abnormalities of placental trophoblast caused by ZIKV infection during early pregnancy compared to 2D cultured cell models. This study deepened the understanding of the effect of ZIKV infection in early pregnancy on placental trophoblast development.

## Methods
### Ethics statement
The experiments with infectious ZIKV were conducted under biosafety level 2 (BSL2) facilities at the Beijing Institute of Microbiology and Epidemiology, AMMS. The experiments conducted in this study using hTSCs were approved by the Ethic Committee of the Center for Reproductive Medicine, Sixth Affiliated Hospital of Sun Yat-Sen University (#2019SZZX-008). The experiments in this study complied with the Guidelines for Stem Cell Research and Clinical Translation issued by International Society for Stem Cell Research (ISSCR) and the Ethical Guidelines for Human Embryonic Stem Cell Research issued by the Ministry of Science and Technology and the Ministry of Health of People's Republic of Chine.

### Culture of human trophoblast stem cells (hTSCs) and cell lines
The hTSCs used in this study included the hTSCs originally established by Wang lab[25], the hTSC strain BT1 gifted by Arima lab[23], and hESC-derived trophoblast-like cells (H1-TS) gifted by Pan lab[69]. The hTSCs, BT1 and H1-TS were cultured following the previously published protocol[25]. In brief, the hTSCs were cultured in 6-well plates pre-coated with 5 μg/ml collagen-IV at 37 °C in 5% CO$_2$. The hTSCs culture medium (hTSM) was prepared as follows: DMEM/F12 medium supplemented with 0.1 mM 2-mercaptoethanol, 0.2% fetal bovine serum (FBS), 0.5% Penicillin-Streptomycin, 0.3% BSA, 1% ITS-X supplement, 0.5 μM A83-01, 2 μM CHIR99021, 1 μM SB431542, 5 μM Y27632, and 0.8 mM VPA. The hTSCs were dissociated with TrypLE Express Enzyme (Gibco) and passaged every 4-5 days at a 1:6 split ratio.

BHK-21 cells and C6/36 cells were purchased from American Type Culture Col lection (ATCC). BHK-21 cells (ATCC, #CCL-10) were cultured in Dulbecco's modified Eagle's medium (Invitrogen) with 10% FBS, 100 U/ml of penicillin, and 100 μg/ml of streptomycin.

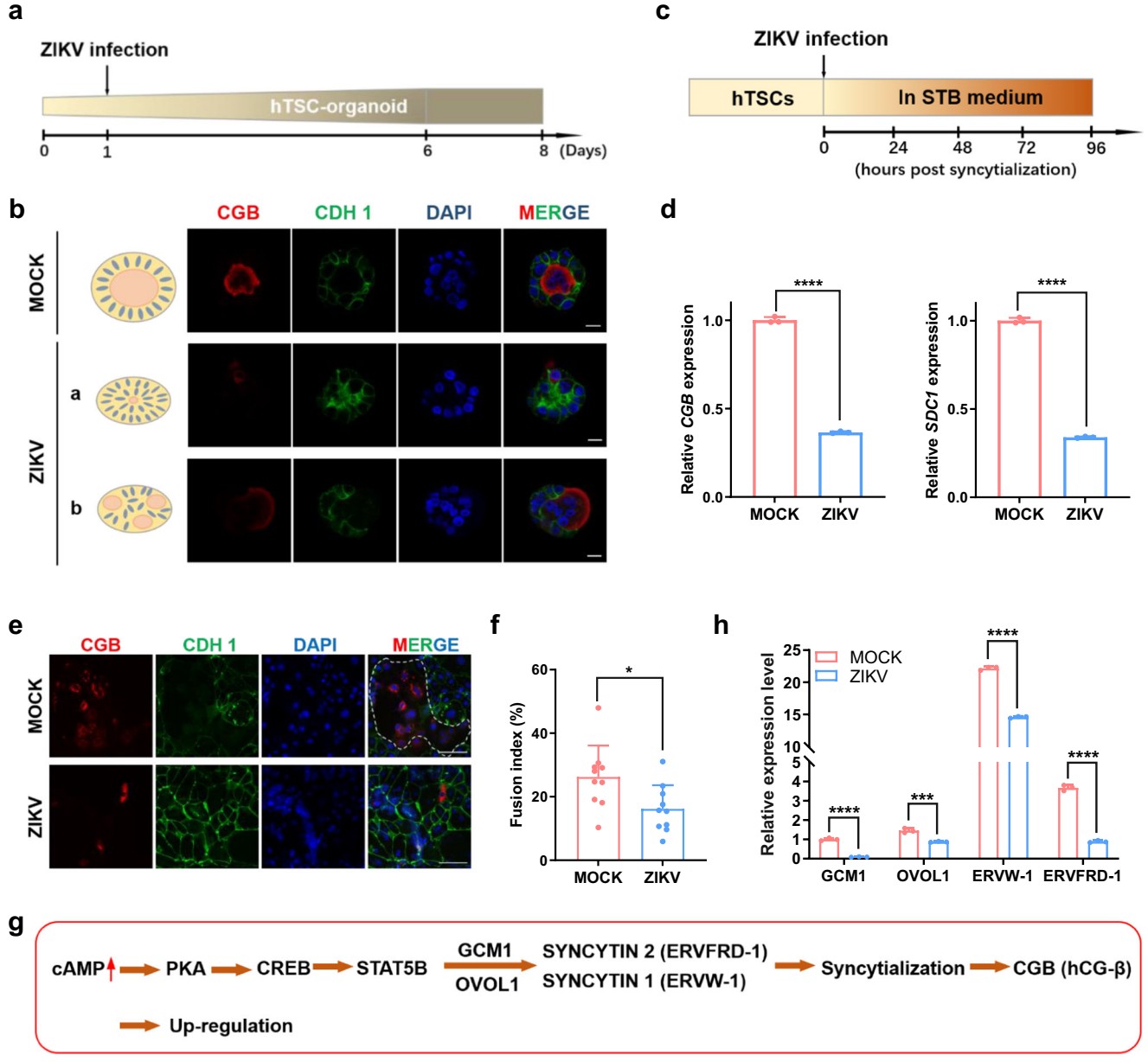

**Fig. 7 | ZIKV infection represses the syncytialization of hTSC-organoids. a** A schematic diagram of the development assay in hTSC-organoids. **b** Representative immunofluorescence images showing the structure of day 8 mock- and ZIKV-infected hTSC-organoids. Nuclei were stained with DAPI. Day 1 hTSC-organoids were exposed to ZIKV at an MOI of 10, and analyzed at day 8. Scale bars: 20 μm. **c** A schematic diagram of the syncytialization assay in hTSCs. **d** Quantification of the expression of the STB markers, CGB and SDC1. The hTSCs were infected with ZIKV at an MOI of 1 and were differentiated into STB^TS. The RNA of the STB^TS derived from mock- and ZIKV-infected hTSCs was collected and analyzed at 96 hours post infection. Two-tailed unpaired t test was used for statistical analysis of significance. *n* = 3 independent experiments. ****, *p* < 0.0001. **e** Representative immuno-fluorescence images of the STB^TS derived from mock- and ZIKV-infected hTSCs.

Immunostaining for CGB and CDH1 was performed at 96 hours after differentiation. Nuclei were stained with DAPI. The white dished line showing the outline of multinucleated STB. Scale bars: 100 μm. **f** Fusion index of the STB^TS derived from mock- and ZIKV-infected hTSCs. Two-tailed unpaired t test was used for statistical analysis of significance. *n* = 10 random views. *, *p* = 0.0199. **g** A schematic diagram showing the regulation of syncytialization by CAMP-PKA-CREB-STAT5B signaling. **h** Quantification of GCM1, OVOL1, ERVW-1 and ERVFRD-1 expression in mock- and ZIKV-infected hTSCs. The hTSCs were exposed to ZIKV at MOIs of 0.1 and 1, and RNA was collected at 48 hours post infection. Two-way ANOVA analysis was used for statistical analysis of significance. *n* = 3 independent experiments. ***, *p* = 0.0004. ****, *p* < 0.0001. Data in this figure are shown as the mean ± s.d.

C6/36 cells (ATCC, #CRL-1660) were cultured in RPMI-1640 with 10% FBS, 100 U/ml of penicillin, and 100 μg/ml of streptomycin.

**Viruses**

The ZIKV GZ01 strain (GenBank accession no: KU820898) was isolated from a Chinese patient returned from Venezuela in 2016. The ZIKV FSS 13025 strain (GenBank accession no: KU955593) was isolated in Cambodia in 2013. Viral stocks were prepared in mosquito C6/36 cells

and titrated in BHK-21 cell by plaque-forming assay. Viral stocks were stored as aliquots at −80 °C.

**Differentiation of human trophoblast stem cells (hTSCs)**

The hTSCs were differentiated into STB by seeding into the 6-well plates pre-coated with 2.5 μg/ml collagen-IV and cultured in STB medium (STM). The STM was prepared as follows: DMEM/F12 medium supplemented with 0.1 mM 2-mercaptoethanol, 0.5% Penicillin-

Streptomycin, 0.3% BSA, 1% ITS-X supplement, 4% KSR, 2.5 μM Y27632, and 2 μM Forskolin. The STM was replaced every other day, and the STB at day 6 was used for analysis.

The hTSCs were differentiated into EVT by seeding into the 6-well plates pre-coated with 1 μg/ml collagen-IV and cultured in EVT medium (ETM). The ETM was prepared as follows: DMEM/F12 medium supplemented with 0.1 mM 2-mercaptoethanol, 0.5% Penicillin-Streptomycin, 0.3% BSA, 1% ITS-X supplement, 4% KSR, 2.5 μM Y27632, 100 ng/ml NRG1, 7.5 μm A83-01, and 2% Matrigel. At day 3, the medium was replaced with ETM2. The ETM2 was prepared as follows: DMEM/F12 medium supplemented with 0.1 mM 2-mercaptoethanol, 0.5% Penicillin-Streptomycin, 0.3% BSA, 1% ITS-X supplement, 4% KSR, 2.5 μM Y27632, 7.5 μm A83-01, and 0.25% Matrigel. At day 5, the medium was replaced with ETM3. The ETM3 was prepared as follows: DMEM/F12 medium supplemented with 0.1 mM 2-mercaptoethanol, 0.5% Penicillin-Streptomycin, 0.3% BSA, 1% ITS-X supplement, 2.5 μM Y27632, 7.5 μm A83-01, and 0.25% Matrigel. The EVT at day 6 was used for analysis.

## Construction of hTSC-organoids and generation of EVT from hTSC-organoids

The hTSCs were dissociated with TrypLE Express Enzyme at 37 °C for 8 min, and $1 \times 10^4$ cells were suspended with 100 μl Matrigel. Drops (12.5 μl) were plated into culture plates. The droplets were set at 37 °C for 15 min and then cultured in hTSM. The medium was replaced every other day. To generate EVT from hTSC-organoids, the medium at day 4 was replaced with ETM and cultured for 2 days. Then, the medium was replaced with NRG-1-removed ETM, and the HLA-G$^+$ EVT appeared after 4 days.

## Growth curves of ZIKV

Growth curves of ZIKV GZ01 and FSS 13025 in hTSCs, STB$^{TS}$ and EVT$^{TS}$ cells were performed in a 24-well plate. Cells were inoculated with ZIKV GZ01 at MOIs of 0.1 or 1 for an hour. The inoculum was removed and washed 3 times with phosphate buffered saline (PBS) to remove unbound viruses before adding fresh medium. Cell supernatants were collected at 12, 24 or 48 hpi. Viral RNA was extracted from the culture supernatant after each passage using the QIAamp Viral RNA Kit (Qiagen) and detected using qPCR.

## Infectivity assay of ZIKV infection on hTSC-organoids

To establish ZIKV infection model in hTSC-organoids, the day 6 hTSC-organoids were incubated with ZIKV at MOIs of 1 or 10 at 37 °C for 12 hours. The inoculum was removed and washed three times with hTSM. The mock- and ZIKV-infected hTSC-organoids were fixed for immunofluorescent staining. To explore the effect of ZIKV infection on the development of hTSC-organoids, the day 1 hTSC-organoids were infected with ZIKV at an MOI of 10 at 37 °C for 12 hours. Then, the inoculum was removed and washed three times with hTSM. The mock- and ZIKV-infected hTSC-organoids were fixed at day 8 for immunofluorescent staining.

## Viral RNA extraction and qPCR

The quantification of viral RNA by qPCR was performed as described previously[70]. Total RNA was isolated using TRIzol reagent (Invitrogen) according to the manufacturer's instructions. Viral RNA was extracted from the culture supernatant using the QIAamp Viral RNA Kit (Qiagen) and detected using qPCR. The determination of the detection limit was based on the lowest level at which viral RNA was detected and remained within the range of linearity of a standard curve (Ct value of 35). qPCR was performed using One Step PrimeScript RT-PCR Kit (Takara, Japan) with the primers and probes described in Table S5. The 20 μl reaction mixtures were set up with 2 μl of RNA. Cycling conditions were as follows: 42 °C for 5 min, 95 °C for 10 s, followed by 40 cycles of 95 °C for 5 s and 60 °C for 20 s.

## ZIKV plaque-forming assay

Purified virus were diluted at four different 10-fold and then were used to infect monolayer of BHK-21 cells at 37 °C for 2 hours. Then, a mix of nutriment solution with agar was coated on the cells. Finally, the cells were incubated with 0.1% crystal violet and the number of plaques was counted.

## Immunofluorescence (IF) staining assay

The hTSC-derived trophoblast cells and hTSC-organoids were fixed with 4% paraformaldehyde (PFA) for 30 min and permeated with 0.5% Triton X-100 for 30 min. Samples were treated with blocking buffer (3% BSA) for an hour, and then incubated in primary antibodies overnight at 4 °C. Anti-ZIKV E protein primary antibody was diluted at 1:1000 (mouse anti-ZIKV E, #GTX133314; rabbit anti-ZIKV E, #GTX133325), and other primary antibodies (anti-GATA 3, #5852; anti-CGB, #ab58310; anti-CDH 1, #3195; anti-Ki67, #9129; anti-HLA-G, #sc-21799; anti-CK 7, #ZM0071) used in this study were diluted at 1:200. Afterward, samples were washed with PBS for three times and treated with secondary antibodies (diluted at 1:200) for an hour at room temperature. Samples were washed for three times in PBS before imaging.

## Fusion index calculation

We calculated the fusion index by staining CDH 1 to show the cell membrane and staining DAPI to label the nuclei. The fusion index was determined by (N-S)/T, where T was the total number of nuclei, S was the number of syncytia, and N was the number of nuclei in the syncytia[71]. The statistical data was shown in Tables S3 & S4.

## Gene knockout in hTSCs using CRISPR/Cas9 system

We designed sgRNA targeting AXL, TIM-1, TYRO3 and MERTK using the website: CRISPick (broadinstitute.org). The sgRNA sequences used in this study were listed in Table S6. The sgRNA was cloned into Lenti-CRISPRv2 vector (Addgene plasmid #52961).

The hTSCs were plated into 24-well plate at a density of $2 \times 10^4$ cells per well, and were infected with lentivirus expressing spCas9 and sgRNAs for 12 hours. Then, the infected hTSCs were treated with 2 μg/ml puromycin for 48 hours. Puromycin resistant clone was screened for genotyping. The genotyping primers used in this study were listed in Table S7.

## RNA extraction and real-time quantitative PCR (qRT-PCR)

Total RNA was extracted using TRIzol reagent. cDNA was synthesized from 1 μg total RNA using HiScript® III All-in-one RT SuperMix. The cDNA (1 μl) was used for qRT-PCR after 2.5-fold dilution of cDNA using Fast SYBR Green Mix Kit. The expression of GAPDH was used as endogenous reference control. Relative expression quantification was performed based on the comparative $C_T$ Method. The primers for AXL, TIM-1, TYRO3, MERTK, GAPDH and C19MC quantification were listed in Table S5. The primers for ISG quantification were listed in Table S8.

## RNA library construction and sequencing

Total RNA from cell or organoid was extracted using TRIzol reagent. Sequencing libraries were generated using NEBNext® UltraTM RNA Library Prep Kit for Illumina® (NEB) following the manufacturer's recommendations and index codes were added to attribute sequences to each sample. Clustering of the index-coded samples was performed on a cBot cluster generation system using a HiSeq PE Cluster Kit v4-cBot-HS (Illumina) according to the manufacturer's instructions. After cluster generation, the libraries were sequenced on the Illumina NovaSeq 6000 platform, and 150-bp paired-end reads were generated. After sequencing, a Perl script was used to filter the original data (raw data) and the clean reads were obtained by removing contaminated reads for adapters and low-quality reads.

### Bulk RNA-sequencing (RNA-Seq) analysis

Clean reads were aligned to the Human genome build 38 (hg38) using Hisat2 v2.1.0. A homemade R script based on Rsubread package (v2.10.5) was used for quantifying the transcript expression levels (counts and estimated TPM). Genes with TPM = 0 across all samples were removed. For each sample, the distribution of gene expression (Log$_2$ TPM) values was plotted and expression thresholds were set according a marker gene (HLA-G), which was the specific expression gene of EVT$^{TS}$. Genes with Log$_2$TPM higher than HLA-G were classified as "expressed". Heatmaps were plotted with the pheatmap package (v1.0.12) in R.

For PCA visualization, estimated read counts were normalized and variance stabilized by regularized log transformation with the rlog() function from the DESeq2 package (v1.36.0). DESeq2 v1.36.0 was used for differential gene expression analysis. Genes with P_adj<0.05, |Log$_2$FoldChanges | >1 and mean base counts>100 were identified as differentially expressed genes (DEGs). Heatmaps were constructed using pheatmap (v1.0.12).

### Single-cell RNA sequencing (scRNA-seq) and data analysis

The Day 8 hTSC-organoids were removed from Matrigel using Cell Recovery Solution at 4 °C for 30 min. The free hTSC-organoids were further digested using TrypLE Express Enzyme in a shaker at 37 °C for 20 min, and pipetted every 5 min. The cells were collected by centrifugation and filtered through a 40 μm Cell Strainer. The cell viability was detected using trypan blue staining.

The construction of transcriptome libraries was performed following the standard protocol of 10x Genomics (v3). Briefly, the cDNA was purified after GEM generation, barcode addition, GEM-RT reaction and bead cleanup. The GEX libraries were generated following the protocol of 10x Genomics Chromium Single Cell 3' Reagent Kits (v3). The concentration of each library was accurately detected for 10x Genomics sequencing.

Raw sequencing data were converted to fastq format using the cellranger mkfastq command (v.7.1.0). scRNA-seq reads were aligned to the GRCh38 (UCSC hg38) reference genome and quantified by cellranger count command using default parameters.

Count data were processed using the R package Seurat (v.4.3.0) (Hao et al., 2021). Gencode v.31 was used for gene identification. Cells with less than 2,000 informative genes expressed were excluded. Count data was log-normalized and scaled to 2,000. scRNA-seq data sets for mock, MOI 1 and MOI 10 samples were integrated using 2000 anchors with reciprocal PCA (RPCA) in Seurat package.

PCA analysis was based on the 2,000 most variable genes and clusters were identified using the Louvain community detection implemented in Seurat's FindClusters function. 2-dimensional representations were generated using uniform manifold approximation and projection (UMAP). For each cluster, the DEGs between mock- and ZIKV (MOI 1 and MOI 10)-infected samples were determined using the FindMarkers function and a Wilcoxon Rank Sum test ('logfc.threshold=0.1, min.pct=0.25'). Gene Ontology (Go) term enrichment analyses were performed with the clusterProfiler package in R. To infer developmental trajectories, Velocity.R (v0.6) package was implemented following the steps in https://rdrr.io/github/satijalab/seurat-wrappers/f/docs/velocity.Rmd.

### Separation of infected cells by flow cytometry assay

Flow cytometry assay was performed as Carlin et al. described[65]. Briefly, the cells were washed with dulbecco's phosphate buffered saline (DPBS) and digested into single cells by TrypLE Express Enzyme. The cells were fixed with 4% PFA for half an hour, and then permeabilized with 0.3% Triton X-100 for half an hour. The cells were blocked for an hour with 3% BSA and then stained with ZIKV E protein antibody for an hour at room temperature. After washing with DPBS for three times, the cells were stained with secondary antibody conjugated with FITC for half an hour at 4 °C. Cells were washed with DPBS for three times and sorted into mock- and ZIKV-infected cells on a FACS cell sorter (BD Biosciences). The uninfected cells were used as negative controls to set the gate.

### Materials availability

All unique reagents and biological materials used in this study are available from the corresponding authors, Dr. Hongmei Wang (wanghm@ioz.ac.cn) and Dr. Cheng-Feng Qin (qincf@bmi.ac.cn), in compliance with Material Transfer Agreements (MTA).

### Statistical analysis

Statistical analysis was performed using GraphPad Prism software (v 9.5) by unpaired t test or two-way ANOVA analysis. Results were shown as the mean only or means ± standard deviation (s.d.). The threshold for statistical significance was $p < 0.05$. n refers to the number of independent replicates of the experiment.

### Reporting summary

Further information on research design is available in the Nature Portfolio Reporting Summary linked to this article.

## Data availability

RNA-seq and scRNA-seq data generated in this study have been deposited in NCBI Gene Expression Omnibus database under accession code GSE229702, which can be viewed via the following link. RNA-seq and scRNA-seq data generated in this study have been deposited in National Genomics Data Center database under accession code HRA002901, which can be viewed via the following link. Source data are provided with this paper.

## Code availability

Our analysis of the data can be viewed on the public repository GitHub (https://doi.org/10.5281/zenodo.8213379) via the following link https://github.com/hyzhou1990/Supp-for-paper-reviewed.

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

## Acknowledgements

Hongmei Wang was supported by grants from the National Key Research and Development Program of China (2018YFE0201103 and 2022YFA1103600) and from National Natural Science Foundation of China (NSFC82192870). Xing-Yao Huang was supported by the Youth Fund from National Natural Science Foundation of China (NSFC82102394). Cheng-Feng Qin was supported by the National Science Fund for Distinguished Young Scholar (NSFC81925025), the Innovative Research Group (NSFC81621005) from the National Natural Science Foundation of China, and the Innovation Fund for Medical Sciences (2019-I2M-5-049) from the Chinese Academy of Medical Sciences. Hang-Yu Zhou was supported by the Natural Science Foundation of Jiangsu Province (BK20220278). We thank Dr. Guangjin Pan for gifting the hESC-derived hTSC-like cells (H1-TS). We thank Dr. Takahiro Arima for gifting the blastocyst-derived hTSC strain (BT1).

## Author contributions

H.M.W. and C.Q. conceived and supervised the project. H.W. constructed gene knockout hTSCs and hTSC-organoids. X.H., M.S., Y.T. and K.L. performed ZIKV infection, viral RNA quantification and plaque-forming experiments. H.W., X.H., Y.W. and B.H. performed immunostaining. H.W. and D.L. performed flow cytometry experiments. H.W., H.Z. and A.W. analyzed all transcriptome data. H.W. and X.H. wrote the manuscript.

## Competing interests

The authors declare no competing interests.
