## [Peer Review File · Nature Communications]

nature portfolio

Peer Review FileReviewer comments, first round

Reviewer #1 (Remarks to the Author):

In this manuscript by Wu et al., the authors have examined the susceptibility of human trophoblast to Zika virus (ZIKV). They demonstrate that trophoblast stem cells are readily infected by ZIKV, but that there is increasing resistance to virus as differentiation to mature lineages proceeds. They also provide evidence that the ZIKV binding proteins AXL & TIM-1 rather than TYRO & MERKT are likely responsible for viral entry. In general, the work appears to have been performed carefully and provides some insights into why the early human conceptus can be infected with ZIKV whereas later stages are more resistant.

This reviewer has two concerns about the study. First, they dismiss the original work of Sheridan et al. (2017) which arrives at essentially the same conclusions as those reported here, on the basis that the model used has "challenged trophoblast identity" (line 83). In fact the criticism of that model has been subject to recent rebuttal. Therefore, the conclusions drawn from the Sheridan et al. paper should be discussed in the light of the present findings.

My second concern is that the organoids, which are derived from trophoblast stem cells" are claimed to recapitulate the "structure, secretory activity etc." of placental villi. In fact, they do not because they are essentially "inside out" with syncytial structure internal rather than surface-exposed and with dividing cells more concentrated on the periphery. In other words, these organoids are not ideal for studying feature of ZIKV infection of villi.

Reviewer #2 (Remarks to the Author):

In this manuscript, Wu et al investigate the detrimental effects of Zika virus (ZIKV) infection in placental cells and elucidated the importance of antiviral immune response for the susceptibility of specific placental cell populations. In summary, they demonstrated that human trophoblast stem cells are modestly more susceptible to ZIKV infection in comparison to syncytiotrophoblast and cytotrophoblast due to differential expression of genes related to the antiviral immune response. They demonstrate that AXL and TIM-1, the canonical receptors for ZIKV, are required for the infection of trophoblast stem cells. Interestingly, the group generated placenta organoids that mimic ZIKV in vivo infection with impaired syncytia formation and increased expression of antiviral immune genes. The results are relevant for the literature of ZIKV congenital infection and clarify some aspects about increased susceptibility of stem cells helping to understand the effects of the infection at early stages of pregnancy.

Major comments

1. The differential gene expression of antiviral genes according to the differentiation state of the cells is interesting. However, the statement about differences in the viral load need to be moderated, since these differences are only 0.3 log, which for viral infections is not significant.
2. According to the differences found in the immunostaining of ZIKV ENV protein, the alterations in viral load seem to be related to viral assembly and protein translation. To clarify this and support the statement that cell differentiation affects the viral load I suggest some assays to quantify infectious viral particles such as plaque forming units or focus forming units.

3. The statistics for most of the graphs need revision (Figures 1, 2 and 4). In the methods section it is stated that an unpaired t-test was performed. However, some results need to be reanalyzed by other tests such as One- or Two-way ANOVA, depending on the number of variables.

4. The differences in ISGs between differentiation stages are interesting and their detrimental role during congenital infection should be discussed. I suggest consideration of Yockey et al (2019) doi: 10.1126/sciimmunol.aao1680.

Minor comments

1. The writing needs revision, There are typos, repeated words, and the grammar requires improvement.

2. Line 136, the statement of a “clear increase” requires statistics between the time points.

3. Authors should indicate the number of experimental replicates for each experiment.

4. Most of the genes presented as ISGs are not ISG, but genes related to the overall innate immune response and other intracellular pathways. Some examples are the genes mentioned in lines 109, 205, 210 and 214.

5. Line 203 requires a reference.

6. RT-PCR needs to be replaced by qPCR.

7. I suggest placing the heatmaps from figure 3B and 3E side by side to make the differences between mock and infected samples clearer.

8. The statistical test performed for each graph should be mentioned in the figure legends.

Reviewer #3 (Remarks to the Author):

Wu et al utilise their newly developed human blastocyst derived trophoblast stem cells as a model system for investigating zika infection. They find that hTSCs are the major target of the virus. Furthermore, that a deficiency of IFN-stimulated genes and the expression of AXL and TIM-1 leave the cells vulnerable to infection. As the cells differentiate, they become more resistant to infection. This study is a valuable addition to the scientific literature as the understanding of the targeting of the zika virus to trophoblast cells and exemplifies the use of trophoblast organoids.

My expertise is in next-generation sequencing, especially single-cell sequencing, and placental biology, so I have restricted my comments to these topics.

Main comments

Overall, this study is well thought out and conducted, however I have some concerns regarding the

hTSCs. Specifically, how do they compare to the other competing trophoblast organoid models?

As with many others in the community, I have concerns over how representative these particular hTSCs are without a detailed comparison. Ideally this would be at the single cell level to ensure specific cell types are compared. The genes identified in this study should be queried in the other competing systems. Others in the field have address this and there have been several recent publications benchmarking the models. Shannon et al, provide online resources to interactively compare the competing methods.

- Cox & Naismith (2022). Here and there a trophoblast, a transcriptional evaluation of trophoblast cell models. <https://doi.org/10.1007/s00018-022-04589-4>

- Shannon et al. (2022) Single-cell assessment of trophoblast stem cell-based organoids as human placenta-modeling platforms.

<https://www.biorxiv.org/content/10.1101/2022.11.02.514970v1.full.pdf>

Do all the competing TSC based organoids lack the IFN-stimulated genes?

Minor comments:

1. What are the metrics for AXL and TIM-1 for them to be picked out from other key genes. Was a particular expression level (TMP), log2fold change used?
2. Any code should be made available on a public repository e.g. GitHub, especially where there are “custom” scripts and modifications to standard pipelines.

Reviewer #4 (Remarks to the Author):

A greater understanding of mechanisms of Zika virus (ZIKV) transplacental transmission is required to develop vaccines and drugs that prevent and treat fetal ZIKV infection. In addition, conflicting results about the susceptibility of human trophoblast cells to ZIKV infection have been reported. This study therefore addresses highly significant questions in the field, as the authors develop models of ZIKV infection in human trophoblast stem cells (hTSCs) and hTSC-derived organoids and investigate ZIKV-host interactions using these new cell culture and organoid models. In particular, the hTSC-derived organoid model of ZIKV infection represents a highly innovative and novel approach for the ZIKV field. The authors used ZIKV strain GZ01 (a clinically relevant ZIKV isolate) to infect hTSCs and hTSC-derived syncytiotrophoblasts (SCT), extravillous trophoblast (EVT), and organoids and perform virologic (viral RNA qRT-PCR and ZIKV E protein antigen IHC) and molecular (CRISPR-Cas9-mediated gene knockout and bulk RNAseq) assays. The authors report that: hTSCs are susceptible to ZIKV infection; AXL and TIM-1 are essential for ZIKV infection of hTSCs; ZIKV induces robust ISG expression in hTSCs but not STB and EVT; and ZIKV disrupts hTSC-organoid structure and inhibits syncytialization. However, the authors’ results are challenging to interpret, as described below.

1) Figure 1: Characterization of ZIKV infection model in hTSCs, STB, and EVT: Only viral RNA copies and E protein expression (IHC) are shown. No difference in viral RNA levels in 12 vs 24 h time points (panel A), low percentage of E antigen-positive cells (panel C), and no data on levels of infectious virus suggest that these cells may not be supporting ZIKV replication. E protein expression may be due to phagocytosed or endocytosed virions, and qRT-PCR detects both + and + ZIKV strand. Moreover, published studies have shown that human trophoblasts can express ZIKV RNA and E

protein without producing infectious viral particles (e.g., PMID: 28776046). Additionally, in panel B, E antigen expression is not different between 24h and 48h, whereas the RNA data in panel A shows higher RNA levels at 48h than 24h. It is important to characterize these new ZIKV infection models using assays that measure infectious virus and additional time points for determining the peak levels. Further, the figure legend states $n = 3$; does this represent 3 different human blastocysts or 3 different experiments using the same human donor-derived cells? It may also be helpful to compare GZ01 infection with another ZIKV strain that has been published.

2) Figure 2: Results showing no effect of AXL and TIM-1 deficiency at 24h after ZIKV infection in panel C are not congruent with data showing effect of AXL or TIM-1 overexpression at 24 h in panel D. Irrespective of the overexpression data, there should be an effect of AXL or TIM-1 deficiency at 24h, as Figure 1 (panel C) shows similar levels of E protein expression in hTSCs at 24h and 48h after ZIKV infection (same MOI = 0.1 was used in both Figs 1 and 2). $n = 3$ human donors or 3 expts using the same donor-derived cells?

Also, published studies with mice deficient in AXL and TIM-1, individually and together, have shown no role for these molecules in ZIKV transplacental transmission (PMID: 28423319). Potential explanations??

3) Figure 3: Published studies have shown that RNAseq analysis of a mixed population of cells containing both ZIKV-infected and uninfected cells impacts both sensitivity and specificity (e.g., PMID: 30206152). As only 10-15% of cells are shown to be infected with ZIKV at MOI = 0.1, most of the signals in the RNAseq data are likely coming from uninfected cells. Also, in general, trophoblasts constitutively produce IFNs—it is therefore critical to examine ISG expression in infected cells separately from uninfected cells in the culture, and IFN production should also be assessed. What percent of cells were infected at MOI = 1? Why was MOI = 1 selected for RNAseq analysis in this figure, whereas MOI = 0.1 was used for experiments in Figures 1 and 2? To directly compare the magnitude and quality of the ISG response following ZIKV infection in TSC vs STB vs EVT cultures, it might be helpful to use cells with similar levels of ZIKV infection (e.g., 10% of cells are infected in each of the 3 cell culture models). Finally, $n = 3$ different human donors or same donor but experimental repeats? Were RNAseq data validated by qRT-PCR or another approach?

4) Figure 4: Figure 1 comments apply here—authors need to demonstrate ZIKV replication by performing an assay that assesses infectious virus production and additional time points after infection. Additional controls, such as a ZIKV isolate other than GZ01 and a closely related virus, such as DENV would also provide rigor, especially since a high MOI = 10 is used in most experiments in this Figure. However, MOI is not mentioned in some of the panels—e.g, in panels C and E, which MOI was used?

5) Figure 5: Since Figure 1 shows that STBs can be infected with ZIKV, it is unclear why CGB and SDC1 expression was not assessed directly in STBs. It might be informative to examine whether CGB, SDC1, and CDH1 expression co-localize in E antigen-positive cells. Rationale for using different MOIs in different panels is also unclear.

Point-by-point response to reviewers:

Reviewer #1 (Remarks to the Author)

In this manuscript by Wu et al., the authors have examined the susceptibility of human trophoblast to Zika virus (ZIKV). They demonstrate that trophoblast stem cells are readily infected by ZIKV, but that there is increasing resistance to virus as differentiation to mature lineages proceeds. They also provide evidence that the ZIKV binding proteins AXL & TIM-1 rather than TYRO3 & MERKT are likely responsible for viral entry. In general, the work appears to have been performed carefully and provides some insights into why the early human conceptus can be infected with ZIKV whereas later stages are more resistant. This reviewer has two concerns about the study.

[Response] We thank the reviewer for recognizing the importance of our study.

1. First, they dismiss the original work of Sheridan et al. (2017) which arrives at essentially the same conclusions as those reported here, on the basis that the model used has "challenged trophoblast identity" (line 83). In fact, the criticism of that model has been subject to recent rebuttal. Therefore, the conclusions drawn from the Sheridan et al. paper should be discussed in the light of the present findings.

[Response] We thank the reviewer for this constructive suggestion. We agree that the study by Sheridan et al. has provided advanced understanding in the vulnerability of human trophoblast cells to ZIKV using primed hESC-differentiated hTSC-like cells, and as suggested we discussed the findings by Sheridan et al. in the discussion section of our revised manuscript, as follows:

Sheridan et al. have shown the vulnerability of human trophoblast cells to ZIKV and evaluated the expression of potential host factors for flavivirus in human trophoblast cells using hESC-derived hTSC-like cells. The hESC-derived hTSC-like cells showed a high expression of TAM family genes (AXL, TYRO3 and MERTK), and low expression of the TIM family gene (TIM-1). However, some differences were indicated between the primary hTSCs and hESC-derived hTSC-like cells, including the differential expression of TIM-1¹, and we confirmed that hESC-derived hTSC-like cells (H1-TS) had much lower TIM-1 expression than blastocyst-derived hTSCs¹ (**Rebuttal Fig. 1**).

Rebuttal Figure 1. Quantification of TIM-1 (HAVCR1) mRNA expression in hTSCs and the hESC (H1)-derived hTSC-like cells (H1-TS). Unpaired *t* test was used for statistical analysis of significance. *n*=3 independent experiments. ****, *p*<0.0001.

Reference:

1. Io, Shingo et al. "Capturing human trophoblast development with naive pluripotent stem cells in vitro." *Cell stem cell* vol. 28,6 (2021): 1023-1039.e13. doi:10.1016/j.stem.2021.03.013

2. My second concern is that the organoids, which are derived from trophoblast stem cells" are claimed to recapitulate the "structure, secretory activity etc." of placental villi. In fact, they do not because they are essentially "inside out" with syncytial structure internal rather than surface-exposed and with dividing cells more concentrated on the periphery. In other words, these organoids are not ideal for studying feature of ZIKV infection of villi.

[Response] We thank the reviewer for raising this concern. We agree with the reviewer that the current trophoblast organoids have an "inside out" structure. We have toned down our claim that "hTSC-organoids recapitulate the structure of original human placental villi" in our revised manuscript. We also agree with the reviewer that the "inside out" organoid is not appropriate for investigating the transplacental infection pathways of pathogens. However, in this study, we focused on the susceptibility and vulnerability of trophoblast cells to ZIKV and we hope the reviewer can agree with us that this model is appropriate for this study.

Reviewer #2 (Remarks to the Author)

In this manuscript, Wu et al. investigate the detrimental effects of Zika virus (ZIKV) infection in placental cells and elucidated the importance of antiviral immune response for the susceptibility of specific placental cell populations. In summary, they demonstrated that human trophoblast stem cells are modestly more susceptible to ZIKV infection in comparison to syncytiotrophoblast and cytotrophoblast due to differential expression of genes related to the antiviral immune response. They demonstrate that AXL and TIM-1, the canonical receptors for ZIKV, are required for the infection of trophoblast stem cells. Interestingly, the group generated placenta organoids that mimic ZIKV in vivo infection with impaired syncytia formation and increased expression of antiviral immune genes. The results are relevant for the literature of ZIKV congenital infection and clarify some aspects about increased susceptibility of stem cells helping to understand the effects of the infection at early stages of pregnancy.

[Response] We thank the reviewer for recognizing that our study clarified some aspects about the increased susceptibility of stem cells and helped to understand the effects of infection at early stages of pregnancy.

Major comments

1. The differential gene expression of antiviral genes according to the differentiation state of the cells is interesting. However, the statement about differences in the viral load need to be moderated, since these differences are only 0.3 log, which for viral infections is not significant.

[Response] We agree with the reviewer that a difference of 0.3 Log is moderate. To further validate the different infectivity of ZIKV to hTSCs, STB^{TS} and EVT^{TS}, in our revised manuscript, we performed the following experiments.

1) We compared the expression of ZIKV RNA in the supernatants of the cells infected with ZIKV at MOIs of 0.1 and 1, and a more significant difference in the peak level of ZIKV RNA in the supernatants between the hTSCs and STB^{TS} and EVT^{TS} was found after infection by ZIKV at an MOI of 1 than that in the cells infected by ZIKV at an MOI of 0.1 (**Rebuttal Fig. 2**), as shown in **Fig. S6** of our revised manuscript.

2) We analyzed the infectious viral particles released from ZIKV-infected hTSCs, STB^{TS} and EVT^{TS} using a plaque-forming assay, which demonstrated that fewer viral particles were produced by STB^{TS} and EVT^{TS} than by hTSCs

after ZIKV infection (**Rebuttal Fig. 3**), as shown in **Fig. 1F** of our revised manuscript.

3) We confirmed our findings by repeating our experiments in another hTSC strain (BT1) and another ZIKV strain FSS 13025 (**Rebuttal Fig. 4**), as shown in **Fig. S1** of our revised manuscript, which showed similar findings to using our hTSCs and ZIKV strain GZ01.

We believe that the above experiments lay a solid foundation for our conclusion that the sensitivity to ZIKV decreased with the differentiation of hTSCs.

Rebuttal Figure 2. Quantification of ZIKV RNA in the supernatants of hTSCs (panel A), STB^{TS} (panel B) and EVT^{TS} (panel C) at 12, 24 and 48 hours post infection. The hTSCs, STB^{TS} and EVT^{TS} were infected with ZIKV at MOIs of 0.1 and 1. Two-way ANOVA analysis was used for statistical analysis of significance. $n=3$ independent experiments. *, $p<0.05$. **, $p<0.01$. ***, $p<0.001$. ****, $p<0.0001$.

Rebuttal Figure 3. Analysis of viral titer in the supernatants of hTSCs, STB^{TS} and EVT^{TS} at 12, 24 and 48 hours post infection by plaque-forming assay. The hTSCs, STB^{TS} and EVT^{TS} were infected with ZIKV at an MOI of 0.1. Two-way ANOVA was used for statistical analysis of significance. $n=3$ independent experiments. *, $p<0.05$. ****, $p<0.0001$. ns, no significance.

Rebuttal Figure 4. Detection of ZIKV RNA in the supernatants of ZIKV-infected hTSC-derived trophoblast cells. (A) Quantification of ZIKV RNA in the supernatants of BT1, STB^{BT1} and EVT^{BT1} at 12, 24 and 48 hours post infection. The BT1, STB^{BT1} and EVT^{BT1} were infected with ZIKV at an MOI of 0.1. Two-way ANOVA analysis was used for statistical analysis of significance. $n=3$ independent experiments. ****, $p<0.0001$. ns, no significance. (B) Quantification of ZIKV RNA in the supernatants of hTSCs, STB^{TS} and EVT^{TS} at 12, 24 and 48 hours post infection. The hTSCs, STB^{TS} and EVT^{TS} were infected with ZIKV strain FSS 13025 at an MOI of 0.1. Two-way ANOVA analysis was used for statistical analysis of significance. $n=3$ independent experiments. *, $p<0.05$. ***, $p<0.001$. ****, $p<0.0001$.

2. According to the differences found in the immunostaining of ZIKV ENV protein, the alterations in viral load seem to be related to viral assembly and protein translation. To clarify this and support the statement that cell differentiation affects the viral load I suggest some assays to quantify infectious viral particles such as plaque forming units or focus forming units.

[Response] We thank the reviewer for this constructive suggestion. As suggested, we performed a plaque-forming assay to confirm production of infectious viral particles by ZIKV-infected hTSCs, STB^{TS} and EVT^{TS} (**Rebuttal Fig. 3**), as shown in **Fig. 1F** of our revised manuscript.

3. The statistics for most of the graphs need revision (Figures 1, 2 and 4). In the methods section it is stated that an unpaired t-test was performed. However, some results need to be reanalyzed by other tests such as One- or Two-way ANOVA, depending on the number of variables.

[Response] We thank the reviewer for this constructive suggestion. We have carefully checked all the statistics and ensured that appropriate statistical methods were used in our revised manuscript. Two-way ANOVA analysis was used in **Figs. 1A, 1F, 2A, 2C, 2D, 7H, S1, S4, S5 and S6** of our revised

manuscript. The statistical methods used for each figure are illustrated in the figure legends of our revised manuscript.

4. The differences in ISGs between differentiation stages are interesting and their detrimental role during congenital infection should be discussed. I suggest consideration of Yockey et al (2019) doi: 10.1126/sciimmunol.aao1680.

[Response] We thank the reviewer for raising this concern. We have discussed the detrimental role of the changed ISGs after ZIKV infection in the result 3 (line 265-267) of our revised manuscript. The genes belonging to type I IFN were elevated after infection, and as Yockey et al. reported, their elevation may disrupt the development of trophoblast cells and the structure of the developing placenta during early embryo development.

Minor comments

1. The writing needs revision, there are typos, repeated words, and the grammar requires improvement.

[Response] We thank the reviewer for raising this concern. We have asked **Nature Research Editing Service** for language editing assistance to optimize our manuscript.

2. Line 136, the statement of a “clear increase” requires statistics between the time points.

[Response] As suggested, we performed a statistical analysis of the changes in ZIKV RNA in the supernatants of ZIKV-infected hTSCs, STB^{TS} and EVT^{TS} at 12, 24 and 48 hpi, demonstrating a clear increase in ZIKV RNA over time, as shown in **Rebuttal Fig. 5**.

This finding was further validated using another hTSC strain BT1 and another ZIKV strain FSS 13025 (**Rebuttal Fig. 4**), as shown in **Fig. S1** of our revised manuscript. We demonstrated the production of infectious viral particles in infected hTSCs, STB^{TS} and EVT^{TS} by the plaque-forming assay (**Rebuttal Fig. 3**), as shown in **Fig. 1F** of our revised manuscript.

Rebuttal Figure 5. Quantification of ZIKV RNA in the supernatants of hTSCs, STB^{TS} and EVT^{TS} at 12, 24 and 48 hours post infection by qPCR. Two-way ANOVA analysis was used for statistical analysis of significance. *n*=3 independent experiments. **, *p*<0.01. ****, *p*<0.0001.

3. Authors should indicate the number of experimental replicates for each experiment.

[Response] We thank the reviewer for raising this concern. We have indicated the number of experimental replicates in each figure legend in our revised manuscript. Each data point used for statistical analysis was collected from at least three independent biological replicates.

4. Most of the genes presented as ISGs are not ISG, but genes related to the overall innate immune response and other intracellular pathways. Some examples are the genes mentioned in lines 109, 205, 210 and 214.

[Response] We thank the reviewer for raising this concern. We agree with the reviewer that our summary of ISGs includes some innate immune genes and intracellular pathways. The ISGs summarized in this study referred to the study published by Wu et al.¹ in Cell, and is broadly defined ISGs. The summary of ISGs is also supported by Schneider et al. and Schoggins et al.^{2,3}. We hope that reviewer can agree with our summary of ISGs.

Reference:

1. Wu, Xianfang et al. "Intrinsic Immunity Shapes Viral Resistance of Stem Cells." *Cell* vol. 172,3 (2018): 423-438.e25. doi:10.1016/j.cell.2017.11.018
2. Schneider, William M et al. "Interferon-stimulated genes: a complex web of host defenses." *Annual review of immunology* vol. 32 (2014): 513-45. doi:10.1146/annurev-immunol-032713-120231
3. Schoggins, John W et al. "A diverse range of gene products are effectors of the type I interferon antiviral response." *Nature* vol. 472,7344 (2011): 481-5.

5. Line 203 requires a reference.

[Response] In our revised manuscript, we have added references where the reviewer indicated.

6. RT-PCR needs to be replaced by qPCR.

[Response] As suggested, in our revised manuscript, we have replaced the RT-PCR with qPCR.

7. I suggest placing the heatmaps from figure 3B and 3E side by side to make the differences between mock and infected samples clearer.

[Response] As suggested, in our revised manuscript, we have placed Fig. 3B and 3E side by side. Furthermore, in revised Fig. 3, we validated the altered ISGs after infection in hTSCs by qRT-PCR, and the result was placed side by side with the panel showing the altered ISGs detected by RNA-seq to ensure easier understanding by the readers.

8. The statistical test performed for each graph should be mentioned in the figure legends.

[Response] In our revised manuscript, we have indicated the statistical methods used in each figure in figure legends.

Reviewer #3 (Remarks to the Author)

Wu et al utilize their newly developed human blastocyst derived trophoblast stem cells as a model system for investigating zika infection. They find that hTSCs are the major target of the virus. Furthermore, that a deficiency of IFN-stimulated genes and the expression of AXL and TIM-1 leave the cells vulnerable to infection. As the cells differentiate, they become more resistant to infection. This study is a valuable addition to the scientific literature as the understanding of the targeting of the zika virus to trophoblast cells and exemplifies the use of trophoblast organoids.

My expertise is in next-generation sequencing, especially single-cell sequencing, and placental biology, so I have restricted my comments to these topics.

[Response] We thank the reviewer for the recognition that our study provides a new understanding of ZIKV targeting and exemplifies the use of trophoblast organoids. As suggested, in **Figs. 5 and 6** of our revised manuscript, we have accessed the transcriptome profile of the hTSC-organoids before and after ZIKV infection at single-cell resolution.

Main comments

1. Overall, this study is well thought out and conducted, however I have some concerns regarding the hTSCs. Specifically, how do they compare to the other competing trophoblast organoid models?

[Response] We thank the reviewer for raising this concern.

1) In response to the reviewer's concern about the findings in this study using hTSCs, we further validated our findings using another hTSC strain (BT1) gifted by Arima lab¹ and another ZIKV strain (FSS 13025), which showed similar findings to the results obtained with our established hTSCs (**Rebuttal Fig. 6**), as shown in **Fig. S1** of our revised manuscript.

Rebuttal Figure 6. Detection of ZIKV RNA in the supernatants of ZIKV-infected hTSC-derived trophoblast cells. (A) Quantification of ZIKV RNA in the supernatants of

BT1, STB^{BT1} and EVT^{BT1} at 12, 24 and 48 hours post infection. The BT1, STB^{BT1} and EVT^{BT1} were infected with ZIKV at an MOI of 0.1. Two-way ANOVA analysis was used for statistical analysis of significance. n=3 independent experiments. ****, p<0.0001. ns, no significance. (B) Quantification of ZIKV RNA in the supernatants of hTSCs, STB^{TS} and EVT^{TS} at 12, 24 and 48 hours post infection. The hTSCs, STB^{TS} and EVT^{TS} were infected with ZIKV strain FSS 13025 at an MOI of 0.1. Two-way ANOVA analysis was used for statistical analysis of significance. n=3 independent experiments. *, p<0.05. ***, p<0.001. ****, p<0.0001.

Reference:

1. Okae, Hiroaki et al. "Derivation of Human Trophoblast Stem Cells." *Cell stem cell* vol. 22,1 (2018): 50-63.e6. doi:10.1016/j.stem.2017.11.004

2) In response to the reviewer's concern about the hTSC-organoids used in this study, currently, there are two main types of trophoblast models simulating the structure and function of placental trophoblast cells: self-organized organoids constructed using Matrigel, and microfluidic-based placenta-on-a-chip. Each has its own strengths in studying viral pathogenicity to the placenta.

I. Self-organized trophoblast organoid:

Self-organized human trophoblast organoids constructed using Matrigel provide a promising approach for studying the development, differentiation and function of trophoblast cells^{1, 2, 3}.

II. Placenta-on-a-chip:

The microphysiological structure of the placenta constructed using microfluidics simulates the structure and function of the placental barrier, which are advantageous for studying maternal-to-fetal material transportation^{4, 5}.

Reference:

1. Turco, Margherita Y et al. "Trophoblast organoids as a model for maternal-fetal interactions during human placentation." *Nature* vol. 564,7735 (2018): 263-267. doi:10.1038/s41586-018-0753-3

2. Haider, Sandra et al. "Self-Renewing Trophoblast Organoids Recapitulate the Developmental Program of the Early Human Placenta." *Stem cell reports* vol. 11,2 (2018): 537-551. doi:10.1016/j.stemcr.2018.07.004

3. Karvas, Rowan M et al. "Stem-cell-derived trophoblast organoids model human placental development and susceptibility to emerging pathogens." *Cell stem cell* vol. 29,5 (2022): 810-825.e8. doi:10.1016/j.stem.2022.04.004

4. Blundell, Cassidy et al. "A microphysiological model of the human placental barrier." *Lab on a chip* vol. 16,16 (2016): 3065-73. doi:10.1039/c6lc00259e

5. Blundell, Cassidy et al. "Placental Drug Transport-on-a-Chip: A Microengineered In

Vitro Model of Transporter-Mediated Drug Efflux in the Human Placental Barrier.
Advanced healthcare materials vol. 7,2 (2018): 10.1002/adhm.201700786.
doi:10.1002/adhm.201700786

We hope that the reviewer can agree with us that our choice of self-organized trophoblast organoids is appropriate for this study.

2. As with many others in the community, I have concerns over how representative these particular hTSCs are without a detailed comparison. Ideally this would be at the single cell level to ensure specific cell types are compared. The genes identified in this study should be queried in the other competing systems. Others in the field have address this and there have been several recent publications benchmarking the models. Shannon et al, provide online resources to interactively compare the competing methods.

- Cox & Naismith (2022). Here and there a trophoblast, a transcriptional evaluation of trophoblast cell models. <https://doi.org/10.1007/s00018-022-04589-4>

- Shannon et al. (2022) Single-cell assessment of trophoblast stem cell-based organoids as human placenta-modeling platforms. <https://www.biorxiv.org/content/10.1101/2022.11.02.514970v1.full.pdf>

Do all the competing TSC based organoids lack the IFN-stimulated genes?

[Response] As suggested, we performed a single-cell RNA sequencing on mock- and ZIKV-infected hTSC-organoids, as shown in **Figs. 5 and 6** of our revised manuscript.

First, we carefully categorized the cell subpopulations in hTSC-organoids into hTSC, CTB, CTB_Fusion and STB, which reflected the differentiation of hTSCs through CTB and CTB_Fusion to mature STB (**Rebuttal Fig. 7**), as shown in **Fig. 5** of our revised manuscript.

Next, we characterized ZIKV-infected trophoblast cells in hTSC-organoids, and identified the infected cells using the immune gene MX1, which is known to be elevated after ZIKV infection. We demonstrated that the proportion of infected cells gradually decreased with CTB differentiation into STB (**Rebuttal Fig. 8**), as shown in **Fig. 6B** of our revised manuscript. The ZIKV-infected hTSCs in hTSC-organoids showed similar changes in the ISG expression as those in ZIKV-infected 2D cultured hTSCs (**Rebuttal Fig. 9**), as shown in **Fig. 6C** of our revised manuscript. We found that the expression of hTSC stemness- and CTB proliferation-related genes was reduced after ZIKV

infection of hTSC-organoids (**Rebuttal Fig. 10**), as shown in **Figs. 6D, 6E, 6F, 6G, 6H and 6I** of our revised manuscript.

Rebuttal Figure 7. Single-cell transcriptome profiles of hTSC-organoids. (A) UMAP showing the cell composition of hTSC-organoids. (B) Dot Plot indicating the expression of trophoblast markers in distinct clusters of hTSC-organoids. The percentage of cells and average gene expression levels that express each gene are presented with differential circle sizes and color intensities, respectively. (C) UMAP overlaid with RNA velocity of hTSC-organoids. Black arrows represent calculated velocity trajectories.

Rebuttal Figure 8. The proportion of MX1-positive cells in CTB, CTB_Fusion and STB from the hTSC-organoids infected with ZIKV at MOIs of 1 and 10.

Rebuttal Figure 9. Dot plot indicating the expression of ISGs in the hTSCs from mock- and ZIKV (MOI 1 and 10)-infected hTSC-organoids. The percentage of cells and average gene expression levels that express each gene are presented with differential circle sizes and color intensities, respectively.

Rebuttal Figure 10. Characterization of ZIKV-infected hTSC-organoids at single-cell level. (A) Dot plot indicating the expression of the stemness-related genes of trophoblast progenitor cells in the hTSCs from mock- and ZIKV (MOI 1 and 10)-infected hTSC-organoids. The percentage of cells and average gene expression levels that express each gene are presented with differential circle sizes and color intensities, respectively. (B) Heatmap showing the differentially expressed genes (Log₂ Fold Changes > 0.1 and Log₂ Fold Changes < -0.1) in the CTB from mock- and ZIKV (MOI 1 and 10)-infected hTSC-organoids. (C) KEGG enrichment of the signaling pathways for up- (left panel) and down (right panel)-regulated genes. (D) Heatmap showing the differentially expressed genes in the CTB_Fusion from mock- and ZIKV (MOI 1 and 10)-infected hTSC-organoids. (E) GO enrichment of the signaling pathways for up- (upper panel) and down (lower panel)-regulated genes. The number of the genes and p value of each gene are presented with differential circle sizes and color intensities, respectively.

In response to the reviewer’s concern on “Do all the competing TSC based organoids lack the IFN-stimulated genes?”. Using scRNA-seq, we demonstrated that important ISGs were expressed at low level in the hTSCs of mock-infected hTSC-organoids, which was consistent with 2D cultured hTSCs (**Rebuttal Fig. 9**), as shown in **Fig. 6C** of our revised manuscript. The findings in this study suggested that the trophoblast progenitor cells in trophoblast organoids constructed using hTSCs lack intrinsic ISG expression.

Minor comments

1. What are the metrics for AXL and TIM-1 for them to be picked out from other key genes. Was a particular expression level (TMP), log2fold change used?

[Response] We thank the reviewer for raising this concern. We probably caused a misunderstanding by the reviewer. We explored the role of four genes of TAM or TIM family in ZIKV infection of hTSCs by gene knockout, but only AXL and TIM-1 showed a significant role. Furthermore, we selected TAM family and TIM family genes to study mainly based on the following reasons:

1) TAM (AXL, MERTK and TYRO3) and TIM (TIM-1) family genes were considered to play potential roles in mediating the entry of some flaviviruses, such as Dengue virus, Ebola virus and Zika virus into certain susceptible cells. TIM-1 has been shown to have a potential role in ZIKV infection of skin cells and endothelial cells. AXL was reported to be a candidate ZIKV entry receptor in radial glial cells, astrocytes, endothelial cells and microglia, while knockout of AXL in neural progenitor cells did not affect their susceptibility to ZIKV, suggesting that AXL may have distinct functions in ZIKV infection of the cells from different tissues. Moreover, the functions of TAM and TIM family genes in ZIKV infection of human trophoblast cells remains unknown, so we investigated their roles in our hTSC-based in vitro models.

2) Our qRT-PCR results showed higher expression of AXL and TIM-1 mRNA in hTSCs than in differentiated STB^{TS} and EVT^{TS}, which is consistent with the trend of decreasing susceptibility to ZIKV with hTSC differentiation. These results suggested that AXL and TIM-1 may have a potential role in mediating ZIKV infection of trophoblast cells.

These results demonstrated the important role of AXL and TIM-1 in ZIKV infection of human trophoblast cells.

2. Any code should be made available on a public repository e.g. GitHub, especially where there are “custom” scripts and modifications to standard pipelines.

[Response] As suggested, in our revised manuscript, our analysis of the data can be viewed on the public repository GitHub via the following link <https://github.com/hyzhou1990/Supp-for-paper-reviewed>.

Reviewer #4 (Remarks to the Author)

A greater understanding of mechanisms of Zika virus (ZIKV) transplacental transmission is required to develop vaccines and drugs that prevent and treat fetal ZIKV infection. In addition, conflicting results about the susceptibility of human trophoblast cells to ZIKV infection have been reported. This study therefore addresses highly significant questions in the field, as the authors develop models of ZIKV infection in human trophoblast stem cells (hTSCs) and hTSC-derived organoids and investigate ZIKV-host interactions using these new cell culture and organoid models. In particular, the hTSC-derived organoid model of ZIKV infection represents a highly innovative and novel approach for the ZIKV field. The authors used ZIKV strain GZ01 (a clinically relevant ZIKV isolate) to infect hTSCs and hTSC-derived syncytiotrophoblasts (SCT), extravillous trophoblast (EVT), and organoids and perform virologic (viral RNA qRT-PCR and ZIKV E protein antigen IHC) and molecular (CRISPR-Cas9-mediated gene knockout and bulk RNAseq) assays. The authors report that: hTSCs are susceptible to ZIKV infection; AXL and TIM-1 are essential for ZIKV infection of hTSCs; ZIKV induces robust ISG expression in hTSCs but not STB and EVT; and ZIKV disrupts hTSC-organoid structure and inhibits syncytialization. However, the authors' results are challenging to interpret, as described below.

[Response] We thank the reviewer for the recognition that our study provides a highly innovative and new approach to the ZIKV field.

1. Figure 1: Characterization of ZIKV infection model in hTSCs, STB, and EVT: Only viral RNA copies and E protein expression (IHC) are shown. No difference in viral RNA levels in 12 vs 24 h time points (panel A), low percentage of E antigen-positive cells (panel C), and no data on levels of infectious virus suggest that these cells may not be supporting ZIKV replication. E protein expression may be due to phagocytosed or endocytosed virions, and qRT-PCR detects both + and + ZIKV strand. Moreover, published studies have shown that human trophoblasts can express ZIKV RNA and E protein without producing infectious viral particles (e.g., PMID: 28776046). Additionally, in panel B, E antigen expression is not different between 24h and 48h, whereas the RNA data in panel A shows higher RNA levels at 48h than 24h. It is important to characterize these new ZIKV infection models using assays that measure infectious virus and additional time points for determining the peak levels.

Further, the figure legend states $n = 3$; does this represent 3 different human blastocysts or 3 different experiments using the same human donor-derived cells? It may also be helpful to compare GZ01 infection with another ZIKV strain that has been published.

[Response] We thank the reviewer for these constructive suggestions. We have performed additional experiments and made the following changes in our revised manuscript.

1) In response to reviewer's concern regarding the infectious viral particles produced by ZIKV-infected hTSCs, STB^{TS} and EVT^{TS}, we performed a ZIKV plaque-forming assay and demonstrated that ZIKV can replicate in hTSCs, STB^{TS} and EVT^{TS} and that the growth rate decreased with hTSC differentiation (**Rebuttal Fig. 11**), as shown in **Fig. 1F** of our revised manuscript.

Rebuttal Figure 11. Analysis of viral titer in the supernatants of hTSCs, STB^{TS} and EVT^{TS} at 12, 24 and 48 hours post infection by plaque-forming assay. The hTSCs, STB^{TS} and EVT^{TS} were infected with ZIKV at an MOI of 0.1. Two-way ANOVA was used for statistical analysis of significance. $n = 3$ independent experiments. *, $p < 0.05$. ****, $p < 0.0001$. ns, no significance.

2) In response to the reviewer's concern that the E protein seems to be not significantly different in the cells between 24 and 48 hours post infection (hpi), we performed immunofluorescence staining for E protein in hTSCs, STB^{TS} and EVT^{TS} infected with ZIKV at an MOI of 1, and significantly more E protein signals were found at 48 hpi than at 24 hpi (**Rebuttal Fig. 12**), as shown in **Fig. S6D, S6E and S6F** of our revised manuscript.

Rebuttal Figure 12. Immunofluorescence staining for Ki67 (a marker of proliferative CTB), CGB (a marker of EVT) and ZIKV E protein in hTSCs (panel A), STB^{TS} (panel B) and EVT^{TS} (panel C). Nuclei were stained with DAPI. The hTSCs, STB^{TS} and EVT^{TS} were infected with ZIKV at an MOI of 1 and analyzed at 24 and 48 hours post infection. The yellow arrow heads indicated the positive intracellular ZIKV E signals in STB^{TS} and EVT^{TS}. Scale bars: 50 μ m.

3) In response to the reviewer's concern on the experimental replication, in our revised manuscript, we have annotated the number of independent experiments in the legends of each figure, which were performed on the same hTSC strain. We further validated the findings using another ZIKV strain BT1 (gifted by Arima lab), and showed similar findings to those using our established hTSCs (**Rebuttal Fig. 13**), as shown in **Fig. S1A** of our revised manuscript.

Rebuttal Figure 13. Quantification of ZIKV RNA in the supernatants of BT1, STB^{BT1} and EVT^{BT1} at 12, 24 and 48 hours post infection. The BT1, STB^{BT1} and EVT^{BT1} were infected with ZIKV at an MOI of 0.1. Two-way ANOVA analysis was used for statistical analysis of significance. $n=3$ independent experiments. ****, $p<0.0001$. ns, no significance.

4) In response to the reviewer's concern about whether the findings obtained in this study can be replicated in other ZIKV strains, we repeated our experiments in another ZIKV strain FSS 13025, which showed similar findings to those using ZIKV strain GZ01 (**Rebuttal Fig. 14**), as shown in **Fig. S1B** of our revised manuscript.

Rebuttal Figure 14. Quantification of ZIKV RNA in the supernatants of hTSCs, STB^{TS} and EVT^{TS} at 12, 24 and 48 hours post infection. The hTSCs, STB^{TS} and EVT^{TS} were infected with ZIKV strain FSS 13025 at an MOI of 0.1. Two-way ANOVA analysis was used for statistical analysis of significance. *n*=3 independent experiments. *, *p*<0.05. ***, *p*<0.001. ****, *p*<0.0001.

2. Figure 2: Results showing no effect of AXL and TIM-1 deficiency at 24h after ZIKV infection in panel C are not congruent with data showing effect of AXL or TIM-1 overexpression at 24 h in panel D. Irrespective of the overexpression data, there should be an effect of AXL or TIM-1 deficiency at 24h, as Figure 1 (panel C) shows similar levels of E protein expression in hTSCs at 24h and 48h after ZIKV infection (same MOI = 0.1 was used in both Figs 1 and 2). *n* = 3 human donors or 3 expts using the same donor-derived cells?

Also, published studies with mice deficient in AXL and TIM-1, individually and together, have shown no role for these molecules in ZIKV transplacental transmission (PMID: 28423319). Potential explanations?

[Response] We thank the reviewer for raising these concerns.

1) In response to reviewer's concern why the viral RNA loads in the supernatants of ZIKV-infected hTSCs with AXL (hTSC^{AXL-/-}) or TIM-1 (hTSC^{TIM-}

^{1-/-}) knockout are not lower than those of ZIKV-infected wildtype hTSCs (hTSC^{WT}) at 24 hours post infection (hpi), the explanations are as follows. As shown in **Fig. 1A**, between the 24 and 48 hpi, the viral RNA loads in the supernatant were still increasing. Although no significant decrease in viral RNA loads in the supernatants of ZIKV-infected hTSC^{AXL-/-} and hTSC^{TIM-1-/-} was observed at 24 hpi, a significant decrease in the growth rate of viral RNA in the supernatant between 24 and 48 hpi was found after knockout of either AXL or TIM-1. We suggest that knockout of either AXL or TIM-1 alone may not be sufficient to result in adequately decreased ZIKV infection of hTSCs that is observable at 24 hpi. The blocking effect of AXL or TIM-1 knockout on ZIKV infection of hTSCs was better demonstrated by the results of immunofluorescence staining. We believe that these results could indicate the important role of either AXL or TIM-1 in ZIKV infection of human trophoblast cells.

2) In response to the reviewer's concern about the experimental replication, "n=3" represents three independent experiments using the same hTSC strain in our study. In our revised manuscript, the findings in this study were further demonstrated using another hTSC strain (BT1) and another ZIKV strain (FSS 13025) (**Rebuttal Figs. 13 and 14**), as shown in **Fig. S1** of our revised manuscript.

3) In response to the reviewer's concern regarding the failure of knockout of Axl and Tim-1 to block the trans-placenta transmission of ZIKV in mice during pregnancy by Hastings et al., Hastings's results are not contradictory to our findings in this study, instead, our findings are a strong addition to the understanding of ZIKV infection of the placenta. On the one hand, Hastings et al. did not evaluate the role of knockout of Axl or Mertk on ZIKV infection of the specific cell types in the placenta, particularly trophoblast cells. On the other hand, the development and structure of the placenta differs between human and mouse, especially in trophoblast cells¹, thus, the phenotypes in mice may not be fully representative of those in humans. In this study, although we have demonstrated the role of AXL and TIM-1 in ZIKV infection of trophoblast cells, their roles in other placental cell types remain unknown. Therefore, whether blocking AXL and TIM-1 can block ZIKV infection of the placenta requires further study.

Reference:

1. Carter, A M. "Animal models of human placentation--a review." *Placenta vol. 28 Suppl A (2007): S41-7. doi:10.1016/j.placenta.2006.11.002*

3. Figure 3: Published studies have shown that RNAseq analysis of a mixed population of cells containing both ZIKV-infected and uninfected cells impacts both sensitivity and specificity (e.g., PMID: 30206152). As only 10-15% of cells are shown to be infected with ZIKV at MOI = 0.1, most of the signals in the RNAseq data are likely coming from uninfected cells. Also, in general, trophoblasts constitutively produce IFNs—it is therefore critical to examine ISG expression in infected cells separately from uninfected cells in the culture, and IFN production should also be assessed. What percent of cells were infected at MOI = 1? Why was MOI = 1 selected for RNAseq analysis in this figure, whereas MOI= 0.1 was used for experiments in Figures 1 and 2? To directly compare the magnitude and quality of the ISG response following ZIKV infection in TSC vs STB vs EVT cultures, it might be helpful to use cells with similar levels of ZIKV infection (e.g., 10% of cells are infected in each of the 3 cell culture models). Finally, n = 3 different human donors or same donor but experimental repeats? Were RNAseq data validated by qRT-PCR or another approach?

[Response] We thank the reviewer for raising these concerns. We have performed additional experiments and made following changes in our revised manuscript.

1) In response to the reviewer's concern about the ISG characterization in hTSCs, STB^{TS} and EVT^{TS}, in our revised manuscript, the intrinsic ISG features of mock-infected hTSCs, STB^{TS} and EVT^{TS} were validated by qRT-PCR (**Rebuttal Fig. 15**), as shown in **Figs. 3C and S5** of our revised manuscript.

Rebuttal Figure 15. (A) Quantification of the expression of highly expressed and lowly expressed ISGs showed by RNA-seq in hTSCs, STB^{TS} and EVT^{TS} by qRT-PCR. $n=3$ independent experiments. (B) Quantification of PNPT1, PABPC4 and CREB3L3 mRNA expression in hTSCs, STB^{TS} and EVT^{TS}. $n=3$ independent experiments. Two-way ANOVA analysis was used for statistical analysis of significance. *, $p<0.05$. **, $p<0.01$. ***, $p<0.0001$. (C) Quantification of NFIL3, ATF3 and RIPK2 mRNA expression in hTSCs, STB^{TS} and EVT^{TS}. $n=3$ independent experiments. Two-way ANOVA analysis was used for statistical analysis of significance. ****, $p<0.0001$. (D) Quantification of MT1G, MT1H, MT1F and MT1X mRNA expression in hTSCs and EVT^{TS}. $n=3$ independent experiments. Two-way ANOVA analysis was used for statistical analysis of significance. *, $p<0.05$. **, $p<0.01$. ****, $p<0.0001$.

2) In response to the reviewer's concern about why the cells infected with ZIKV at an MOI of 1 were used for RNA-seq analysis, considering the low proportion of infected cells at an MOI of 0.1, we showed that the proportion of infected cells was significantly increased after infection at an MOI of 1 (**Rebuttal Figs. 16 and 17**), as shown in **Fig. S6** of our revised manuscript. Thus, we choose to detect the changes of ISGs in the hTSCs, STB^{TS}, and EVT^{TS} infected with ZIKV at an MOI of 1.

Rebuttal Figure 16. Quantification of ZIKV RNA in the supernatants of hTSCs (panel A), STB^{TS} (panel B) and EVT^{TS} (panel C) at 12, 24 and 48 hours post infection. The hTSCs, STB^{TS} and EVT^{TS} were infected with ZIKV at MOIs of 0.1 and 1. Two-way ANOVA analysis was used for statistical analysis of significance. *n*=3 independent experiments. *, *p*<0.05. **, *p*<0.01. ***, *p*<0.001. ****, *p*<0.0001.

Rebuttal Figure 17. Immunofluorescence staining for Ki67 (a marker of proliferative CTB), CGB (a marker of STB) and ZIKV E protein in hTSCs (panel A), STB^{TS} (panel B) and EVT^{TS} (panel C). Nuclei were stained with DAPI. The hTSCs, STB^{TS} and EVT^{TS} were infected with ZIKV at an MOI of 1 and analyzed at 24 and 48 hours post infection. The yellow arrow heads indicated the positive intracellular ZIKV E signals in STB^{TS} and EVT^{TS}. Scale bars: 50 μm.

3) In response to the reviewer's suggestion that the infected cells should be separated from the mixed cells for validation of changed ISGs following ZIKV infection, in our revised manuscript, we examined the proportions of infected cells after ZIKV infection among hTSCs, STB^{TS} and EVT^{TS} at an MOI of 1 by FACS according to the method published by Carlin et al.¹, which showed decreased sensitivity to ZIKV infection with hTSC differentiation (**Rebuttal Fig. 18**), as shown in **Fig. S7** of our revised manuscript. Furthermore, we separated the ZIKV-infected hTSCs by FACS and verified the changes in ISGs shown by RNA-seq after infection using qRT-PCR (**Rebuttal Fig. 19**), as shown in **Fig. 3G** of our revised manuscript. The qRT-PCR results showed similar changes in ISGs to the RNA-seq results.

Rebuttal Figure 18. FACS analysis of ZIKV-infected hTSCs (panel A), STB^{TS} (panel B) and EVT^{TS} (panel C) using ZIKV E protein antibody. The hTSCs, STB^{TS} and EVT^{TS} were infected with ZIKV at an MOI of 1 and analyzed at 48 hours post infection.

Rebuttal Figure 19. Quantification of the relative expression of ISGs in ZIKV-infected hTSCs compared to that in mock-infected hTSCs by qRT-PCR. The hTSCs were infected with ZIKV at an MOI of 1 and the infected cells were separated by FACS for qRT-PCR analysis at 48 hours post infection. *n*=3 independent experiments.

4) To further validate the changes in ISGs in hTSCs after ZIKV infection, we analyzed the effect of ZIKV infection on trophoblast cells at single-cell resolution, as shown in **Figs. 5 and 6** of our revised manuscript. We identified the infected hTSCs in hTSC-organoids, and demonstrated that the changes in ISGs in the infected hTSCs of hTSC-organoids were consistent with those in ZIKV-infected 2D cultured hTSCs (**Rebuttal Fig. 20**), as shown in **Fig. 6C** of our revised manuscript.

Rebuttal Figure 20. Dot plot indicating the expression of ISGs in the hTSC from mock- and ZIKV (MOI 1 and 10)-infected hTSC-organoids. The percentage of cells and average gene expression levels that express each gene are presented with differential circle sizes and color intensities, respectively.

Reference

1. Carlin, Aaron F et al. "Deconvolution of pro- and antiviral genomic responses in Zika virus-infected and bystander macrophages." *Proceedings of the National Academy of Sciences of the United States of America* vol. 115,39 (2018): E9172-E9181. doi:10.1073/pnas.1807690115

4. Figure 4: Figure 1 comments apply here—authors need to demonstrate ZIKV replication by performing an assay that assesses infectious virus production and additional time points after infection. Additional controls, such as a ZIKV isolate other than GZ01 and a closely related virus, such as DENV would also provide rigor, especially since a high MOI = 10 is used in most experiments in this Figure. However, MOI is not mentioned in some of the panels—e.g, in panels C and E, which MOI was used?

[Response] We thank the reviewer for raising these concerns. We have performed additional experiments and made the following changes in our revised manuscript.

1) In response to the reviewer's concern about the ability of infected cells to produce infectious viral particles, as suggested, we have verified the production of infectious viral particles by ZIKV-infected hTSCs, STB^{TS} and EVT^{TS} using plaque-forming assay (**Rebuttal Fig. 11**), as shown in **Fig. 1F** of our revised manuscript. We demonstrated similar findings using another hTSC strain, BT1, and another ZIKV strain, FSS 13025, which showed decreased sensitivity to ZIKV with hTSC differentiation (**Rebuttal Figs. 13 and 14**), as shown in **Fig. S1** of our revised manuscript. We believe that these results lead to the conclusion that the resistance to ZIKV increases with hTSC differentiation.

2) In response to the reviewer's concern why an MOI of 10 was used for ZIKV-infection of hTSC-organoids, we found that ZIKV mainly infects the mononucleated CTB of hTSC-organoids when infected at an MOI of 1, while

both CTB and STB can be infected when infected at an MOI of 10. Thus, an MOI of 10 was used to construct a ZIKV infection model in hTSC-organoids.

3) In response to the reviewer's concern that the MOIs used in some panels are unclear, in our revised manuscript, we have indicated the MOIs used for ZIKV infection in the legend of each figure.

5. Figure 5: Since Figure 1 shows that STBs can be infected with ZIKV, it is unclear why CGB and SDC1 expression was not assessed directly in STBs. It might be informative to examine whether CGB, SDC1, and CDH1 expression co-localize in E antigen-positive cells. Rationale for using different MOIs in different panels is also unclear.

[Response] We thank the reviewer for raising these concerns.

1) In response to the reviewer's concern on whether there is co-localization of the STB marker genes and ZIKV E protein in this figure, we performed IF staining for ZIKV E protein and CDH 1 on the STB differentiated from infected hTSCs at 96 hpi, and demonstrated that ZIKV-infected hTSCs rarely undergo syncytialization by immunofluorescence staining for ZIKV E protein (**Rebuttal Fig. 21**), as shown in **Fig. S11** of our revised manuscript. We have indicated the MOIs used for ZIKV infection in our revised legends of each figure.

Rebuttal Figure 21. Immunofluorescence staining for ZIKV and CDH 1 in STB^{TS}. The hTSCs were infected with ZIKV at an MOI of 1 and induced to STB^{TS} and analyzed at 96 hours. Scale bars: 100 μ m.

2) In response to the reviewer's concern about the unclear MOIs used in some panels, in revised manuscript, we have annotated the MOIs used for each panel in the legends.

Reviewer comments, second round

Reviewer #2 (Remarks to the Author):

I am reasonably satisfied with the authors response. No further comments.

Reviewer #3 (Remarks to the Author):

Specifically, how do the hTSCs compare to the other competing trophoblast organoid models
The authors use an additional strain of hTSCs (BT1) to address the concern that the original hTSCs used may differ from the other competing hTSC models in the literature. This is a great addition, and does alleviate some of my concerns as the results replicate in this other strain. However, I would have liked to see a simple comparison of gene expression for some key genes e.g ISG between some of the other published hTSC models and the ones used in the paper. This is irrespective of Zika infection, and purely to see that the chosen hTSCs are indeed equivalent to competing ones, for example any of the five listed by the authors in the rebuttal. The bioRxiv paper by Shannon et al, provide an online resource for querying gene expression in competing hTSCs. For example, it would be possible to quickly replicate the dotplot in Rebuttal figure 7 for competing hTSC models, and also for ISGs calculated from the authors data and compared to the competing models. I would suggest this might make a supplemental figure to confirm the hTSCs used are indeed representative.

Do all the competing TSC based organoids lack the IFN-stimulated genes?

The addition of the single cell sequencing really strengthens the conclusions of the paper and allows interrogation of the zika virus infection in a cell type specific manner.

Any code should be made available on a public repository e.g. GitHub, especially where there are "custom" scripts and modifications to standard pipelines.

I was pleased to see that the code is now uploaded and available on GitHub to help reproducibility and replication of the findings in the paper.

Reviewer #4 (Remarks to the Author):

The authors have done a great job revising this manuscript. They performed several new experiments and addressed almost all my comments.

The only remaining concern is that authors do not compare/contrast IFN and ISG expression in infected vs uninfected hTSCs in the same culture--this is important because only 50-60% of cells appear to be infected in the hTSC cell culture model, comparison of both cell populations in the same culture allows for identification of virally-regulated pathways in hTSCs.

Alternatively, have the authors performed single cell RNA-seq in a manner that allows for identification of virally-infected vs uninfected cells?

Point-by-point response to reviewers:

Reviewer #3 (Remarks to the Author)

Specifically, how do the hTSCs compare to the other competing trophoblast organoid models

The authors use an additional strain of hTSCs (BT1) to address the concern that the original hTSCs used may differ from the other competing hTSC models in the literature. This is a great addition, and does alleviate some of my concerns as the results replicate in this other strain. However, I would have liked to see a simple comparison of gene expression for some key genes e.g ISG between some of the other published hTSC models and the ones used in the paper. This is irrespective of Zika infection, and purely to see that the chosen hTSCs are indeed equivalent to competing ones, for example any of the five listed by the authors in the rebuttal. The bioRxiv paper by Shannon et al, provide an online resource for querying gene expression in competing hTSCs. For example, it would be possible to quickly replicate the dotplot in Rebuttal figure 7 for competing hTSC models, and also for ISGs calculated from the authors data and compared to the competing models. I would suggest this might make a supplemental figure to confirm the hTSCs used are indeed representative.

[Response] We thank the reviewer for recognizing the importance of the additional experiments in the first round revision.

As suggested by Review #3, to demonstrate that the hTSCs used in this study is representative, we compared the expression characteristics of key genes, including ISGs and trophoblast marker genes, in the transcriptome data of our hTSC-derived trophoblast cells with the scRNA-seq data of competing models published by Shannon et al, which showed that their expression is conserved in hTSCs of different origins, and we have added two supplementary figures (**Figs. S6 and S12**) in second revised manuscript. Here are the detailed responses.

As shown in Fig. S6 (**Rebuttal Fig. 1**), ISGs showed a similar expression profile in our hTSC-derived trophoblast cells with the trophoblast cells in competing models, and the trophoblast cells in competing models also lacked the expression of representative antiviral ISGs.

As shown in Fig. S12 (**Rebuttal Fig. 2**), we demonstrated that the trophoblast cells in our hTSC-organoids had similar expression profile of trophoblast marker genes with those in competing models.

Taken together, we demonstrated that the hTSCs used in this study are

indeed representative.

Rebuttal Figure 1. The relative expression of ISGs in hTSCs/CTB, STB^{TS}/SCT_p and EVT^{TS}/EVT of our and competing models.

Heatmap showing the relative expression of ISGs with high (up) and low (bottom) expression in our hTSCs, STB^{TS} and EVT^{TS} (left panel), and in CTB, SCT_p and EVT of competing models (right panel).

Rebuttal Figure 2. The expression of trophoblast marker genes in the trophoblast

cells of competing models.

Dot plot indicating the expression of trophoblast marker genes in the scRNA-seq data of competing trophoblast organoids. SCTp, syncytiotrophoblast precursors. cCTB, column CTB-like cell. TSC, trophoblast stem cell.

Do all the competing TSC based organoids lack the IFN-stimulated genes?

The addition of the single cell sequencing really strengthens the conclusions of the paper and allows interrogation of the zika virus infection in a cell type specific manner.

Any code should be made available on a public repository e.g. GitHub, especially where there are “custom” scripts and modifications to standard pipelines.

I was pleased to see that the code is now uploaded and available on GitHub to help reproducibility and replication of the findings in the paper.

[Response] We thank the reviewer for his/her well recognition of our first revision.

Reviewer #4 (Remarks to the Author)

The authors have done a great job revising this manuscript. They performed several new experiments and addressed almost all my comments.

The only remaining concern is that authors do not compare/contrast IFN and ISG expression in infected vs uninfected hTSCs in the same culture--this is important because only 50-60% of cells appear to be infected in the hTSC cell culture model, comparison of both cell populations in the same culture allows for identification of virally-regulated pathways in hTSCs.

Alternatively, have the authors performed single cell RNA-seq in a manner that allows for identification of virally-infected vs uninfected cells?

[Response] We thank the reviewer for recognizing our additional experiments performed in the first round of revision.

As suggested by Reviewer #4, to profile IFN expression in trophoblast cells, we analyzed IFN expression in the transcriptome data of infected and uninfected 2D cultured hTSC-derived trophoblast cells and the scRNA-seq data from ZIKV-infected hTSC-organoids, which showed that trophoblast cells lacked intrinsic expression of representative antiviral IFNs as well, while some essential antiviral IFNs, such as IFNL1 and IFNL3, were activated after infection, and we have added two supplementary figures (**Figs. S9 and S13C**) in second revised manuscript. Here are the detailed responses.

1) In response to reviewer’s concern about comparing IFN expression in

infected and uninfected cells, we compared IFN expression in infected and uninfected 2D cultured cells and scRNA-seq data from ZIKV-infected hTSC-organoids, and added Figs. S9 and S13C in second revised manuscript.

As shown in Fig. S9A (**Rebuttal Fig. 3**), we analyzed the intrinsic expression of major IFN family genes, including IFN α , IFN β , IFN γ , IFN δ , IFN ϵ , IFN λ 1, IFN λ 3, IFN τ , IFN ω , in hTSCs, STB^{TS} and EVT^{TS}. The list of IFN family genes was collected from National Center for Biotechnology Information (NCBI) database. The expression level of HLA-G in hTSCs was used as a criterion to evaluate the intrinsic expression of IFNs in hTSC-derived trophoblast cells. We found that only few IFNs were intrinsically expressed in trophoblast cells, and no representative antiviral IFNs were intrinsically expressed in trophoblast cells. As shown in Fig. S9B (**Rebuttal Fig. 4**), we further analyzed IFN expression in infected trophoblast cells, and we found that apart from the genes also involved in ISGs, essential antiviral IFNs, such as IFNL1 and IFNL3, were activated in all types of trophoblast cells after ZIKV infection at an MOI of 1. Since hTSCs were the main trophoblast cell type for ZIKV infection, infected hTSCs showed most activated IFNs compared to STB^{TS} and EVT^{TS}.

To further validate the activated IFNs in infected hTSCs, we further analyzed their expression in MX1-postived hTSCs in ZIKV-infected hTSC-organoids. As shown in Fig. 13C (**Rebuttal Fig. 5**), we demonstrated that important antiviral IFNs were indeed activated in infected hTSCs.

Taken together, we demonstrated that trophoblast cells also lack intrinsic expression of representative antiviral IFNs, and antiviral IFNs were activated after ZIKV infection.

Rebuttal Figure 3. Heatmap showing the z-score TPM of IFNs in hTSCs, STBT^{TS} and EVT^{TS}.

Rebuttal Figure 4. Heatmap showing changed IFNs in ZIKV-infected hTSCs, STBT^{TS} and EVT^{TS}.

Rebuttal Figure 5. Dot plot indicating the expression of the IFNs in the hTSCs from mock- and ZIKV (MOI 1 and 10)-infected hTSC-organoids.

In summary, we have compared the IFN and ISG expression in infected and uninfected cell populations in the same culture as suggested by the reviewer. The reviewer also mentioned that “Alternatively, have the authors performed single cell RNA-seq in a manner that allows for identification of virally-infected vs uninfected cells?”. We totally understand and agree with the reviewer that this is an important issue, and the knowledge gained from such experiments will better contribute to the understanding of the effects of viral infection on cells. To do this, we need to separate infected and uninfected cells and the only available method is FACS. However, FACS means the cells need to be fixed, permeabilized and stained, which greatly impairs the quality of RNA and the RNA is not qualified for RNA-seq. In the future, if better strategies to separate infected and uninfected cells are developed, a scRNA-seq study on this will definitely provide a better insight into virus-host interaction. We hope that the reviewer is satisfied with our additional experiments and explanations. Thank you very much again for all your great comments and suggestions during the two rounds of revision.

Reviewer comments, third round

Reviewer #3 (Remarks to the Author):

I am satisfied with the new data comparing TSC models, to show that are broadly representative. Ideally the heatmaps will be provided as supplementary information with the paper

Reviewer #4 (Remarks to the Author):

The authors argue that existing RNA-seq approaches (both single cell and bulk) do not discriminate the response of infected vs uninfected cells in the same culture. This is incorrect. Zanini et al (PMID 29451494 and PMID 30530648) have developed virus-inclusive single cell RNA-seq to interrogate the cellular response of flavivirus-infected vs uninfected cells, and Carlin et al (PMID 30206152) and Branche et al (PMID 36097162) have published bulk RNA-seq analysis of ZIKV-infected vs uninfected cells from the same culture. It is important to state the caveats of studying the cellular response to flaviviral infections using approaches that do not separate infected vs uninfected cells. Carlin et al (PMID 30206152) reported that over 600 and 300 genes, respectively, are assigned to be falsely upregulated and downregulated in a culture containing 36% of infected cells.

Point-by-point response to reviewers:

Reviewer #3 (Remarks to the Author)

I am satisfied with the new data comparing TSC models, to show that are broadly representative. Ideally the heatmaps will be provided as supplementary information with the paper.

[Response] We thank the reviewer for recognizing the additional experiments in second round revision, and all the additional figures in second round rebuttal have been included in the supplementary information of final revised manuscript.

Reviewer #4 (Remarks to the Author)

The authors argue that existing RNA-seq approaches (both single cell and bulk) do not discriminate the response of infected vs uninfected cells in the same culture. This is incorrect. Zanini et al (PMID 29451494 and PMID 30530648) have developed virus-inclusive single cell RNA-seq to interrogate the cellular response of flavivirus-infected vs uninfected cells, and Carlin et al (PMID 30206152) and Branche et al (PMID 36097162) have published bulk RNA-seq analysis of ZIKV-infected vs uninfected cells from the same culture. It is important to state the caveats of studying the cellular response to flaviviral infections using approaches that do not separate infected vs uninfected cells. Carlin et al (PMID 30206152) reported that over 600 and 300 genes, respectively, are assigned to be falsely upregulated and downregulated in a culture containing 36% of infected cells.

[Response] We thank the reviewer for providing new knowledge on the use of virus-inclusive scRNA-seq (viscRNA-seq) in virology research, and we agree that it is helpful for studying pathogen-host interaction. Accordingly, we have added an outlook on the advantages of applying viscRNA-seq to the study of pathogen-host interactions to the discussion section of the final revised manuscript, as follows 'In this study, the host immune response of trophoblast cells to ZIKV infection was obtained using bulk RNA-seq and validated using qRT-PCR after sorting out infected and uninfected cells by FACS, which accuracy will be further improved by recently developed virus-inclusive scRNA-seq techniques (Branche et al., 2022; Carlin et al., 2018; Zanini et al., 2018a; Zanini et al., 2018b).' And also, in our future study, we will attempt to establish such methods for studying pathogen-host interactions more accurately. Thanks again for the valuable suggestions!